# The impact of weather pattern and related transport processes on aviation's contribution to ozone and methane concentrations from $NO_x$ emissions

Simon Rosanka[1,a], Christine Frömming[2], and Volker Grewe[1,2]

[1]Delft University of Technology, Faculty of Aerospace Engineering, Section Aircraft Noise Climate Effects, Delft, The Netherlands
[2]Deutsches Zentrum für Luft- und Raumfahrt, Institute of Atmospheric Physics, Oberpfaffenhofen, Germany
[a]now at: Forschungszentrum Jülich GmbH, Institute of Energy and Climate Research, IEK-8: Troposphere, Jülich, Germany

**Correspondence:** Volker Grewe (volker.grewe@dlr.de)

**Abstract.** Aviation attributed climate impact depends on a combination of composition changes in trace gases due to emissions of carbon dioxide ($CO_2$) and non-$CO_2$ species. Nitrogen oxides ($NO_x = NO + NO_2$) emissions induce an increase in ozone ($O_3$) and a depletion of methane ($CH_4$) leading to a climate warming and a cooling, respectively. In contrast to $CO_2$, non-$CO_2$ contributions to the atmospheric composition are short lived and are thus characterised by a high spatial and temporal
variability. In this study, we investigate the influence of weather patterns and their related transport processes on composition changes caused by aviation attributed $NO_x$ emissions. This is achieved by using the atmospheric chemistry model EMAC (ECHAM/MESSy). Representative weather situations were simulated in which unit $NO_x$ emissions are initialised in specific air parcels at typical flight altitudes over the North Atlantic flight sector. By explicitly calculating contributions to the $O_3$ and $CH_4$ concentrations induced by these emissions, interactions between trace gas composition changes and weather conditions
along the trajectory of each air parcel are investigated. Previous studies showed a clear correlation between the prevailing weather situation at the time when the $NO_x$ emission occurs and the climate impact of the $NO_x$ emission. Here, we show that the aviation $NO_x$ contribution to ozone is characterised by the time and magnitude of its maximum and demonstrate that a high $O_3$ maximum is only possible if the maximum occurs early after the emission. Early maxima occur only if the air parcel, in which the $NO_x$ emission occurred, is transported to lower altitudes, where the chemical activity is high. This
downward transport is caused by subsidence in high pressure systems. A high ozone magnitude only occurs if the air parcel is transported downward into a region in which the ozone production is efficient. This efficiency is limited by atmospheric $NO_x$ and $HO_x$ concentrations during summer and winter, respectively. We show that a large $CH_4$ depletion is only possible if a strong formation of $O_3$ occurs due to the $NO_x$ emission and if high atmospheric $H_2O$ concentrations are present along the air parcel's trajectory. Only air parcels, which are transported into tropical areas due to high pressure systems, experience
high concentrations of $H_2O$ and thus a large $CH_4$ depletion. Avoiding climate sensitive areas by re-routing aircraft flight tracks is currently computationally not feasible due to the long chemical simulations needed. The findings of this study form a basis of a better understanding of $NO_x$-climate sensitive areas and by this will allow to propose an alternative approach to estimate aviation's climate impact on a day-to-day basis, based on computationally cheaper meteorological simulations

without computationally expensive chemistry. This comprises a step towards a climate impact assessment of individual flights, here with the contribution of aviation $NO_x$ emissions to climate change, ultimately enabling routings with a lower climate impact by avoiding climate-sensitive regions.

## 1 Introduction

The importance of anthropogenic climate change is well established since years (Shine et al., 1990) and it is well known that air traffic contributes substantially to the total anthropogenic climate change (Lee et al., 2009; Brasseur et al., 2016; Grewe et al., 2017a). A major fraction of its contribution comes from non-$CO_2$ emissions, which lead to changes in greenhouse gas concentrations as well as contrail and contrail-cirrus formation in the atmosphere (Kärcher, 2018). The climate impact of $CO_2$ is mainly characterised by the emissions strength, due to its long lifetime. However, non-$CO_2$ effects are known to be
characterised by a high spatial and temporal variability. This implies that the total contribution to concentrations of non-$CO_2$ emissions is not only influenced by the emissions strength but also by the time and location of the emission itself.

    Nitrogen oxides ($NO_x = NO + NO_2$) lead to a formation of ozone ($O_3$) following a catalytic reaction. NO reacts with $HO_2$ forming $NO_2$. Via photodissociation, $NO_2$ forms $O(^3P)$ leading to the formation of $O_3$.

$$HO_2 + NO \quad \rightarrow \quad OH + NO_2 \tag{R1}$$
$$NO_2 + h\nu(\lambda \leq 410nm) \quad \rightarrow \quad NO + O(^3P) \tag{R2}$$
$$O(^3P) + O_2 + M \quad \rightarrow \quad O_3 + M \tag{R3}$$

The additionally formed OH leads to an oxidation of $CH_4$:

$$CH_4 + OH \rightarrow CH_3 + H_2O \tag{R4}$$

The destruction of $CH_4$ leads to a reduced $CH_4$ lifetime and a change in $HO_x$ ($HO_x=OH+HO_2$) towards higher OH concen-
trations. This leads to a reduced $O_3$ production, known as primary mode ozone (PMO, Wild et al. (2001)). Compared to the short-term increase in $O_3$, PMO has a long lifetime. An earlier study by Wild et al. (2001) demonstrated that the initial positive climate impact is gradually reduced by PMO leading to a negative climate impact after about 24 years. Earlier studies already identified that the climate impact resulting from aviation attributed $NO_x$ emissions varies strongly within the atmosphere. In general, the increase in $O_3$ has a warming effect whereas the depletion of $CH_4$ leads to a reduced warming, i.e. net-cooling.
The warming caused by $O_3$ is higher than the cooling via $CH_4$ leading to an overall warming due to aviation attributed $NO_x$ emissions (Lee et al., 2009; Grewe et al., 2019). Köhler et al. (2013) showed that the climate impact is larger for emissions occurring in lower than in higher latitudes. A larger climate impact also occurs in regions with low aviation activity for the same amount of $NO_x$. For example, the resulting climate impact for the same amount of aviation $NO_x$ emission is higher in India (low aviation activity) than in Europe (high aviation activity). A similar impact was identified by Stevenson and Derwent
(2009). The general lower background $NO_x$ concentration in the Southern Hemisphere (SH) also explains the inter-hemispheric difference of the resulting climate impact from $NO_x$ emissions. In the SH the climate impact from the same amount of $NO_x$

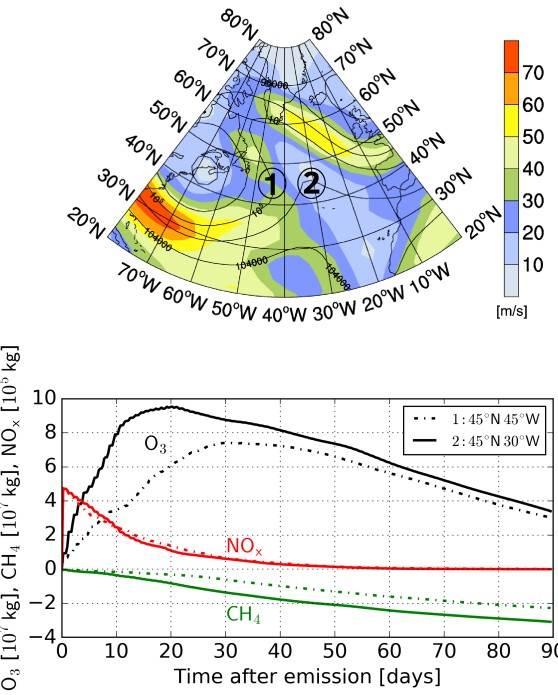

**Figure 1.** Top: Weather conditions at time of emission represented by geopotential height (black contours in gpm) and wind velocities (see colorbar, in $\mathrm{m\,s^{-1}}$) at 250 hPa. Bottom: The global composition changes in $O_3$ and $CH_4$ induced by the emitted $NO_x$ at two emission locations.

is generally larger. Köhler et al. (2008) identified that the emission altitude strongly influences the resulting climate impact, which is generally larger for emissions at high altitudes. Frömming et al. (2012) demonstrated that the overall climate impact can be reduced by adapting flight altitudes, suggesting a possible mitigation strategy. The season in which the emission occurs also influences the resulting climate impact. Gilmore et al. (2013) identified that the production of $O_3$ is about 50 % higher and 40 % lower in summer and winter, respectively, when compared to the annual mean. Grewe et al. (2017a) and Frömming et al. (2020) demonstrated that the total change in ozone is larger if the $NO_x$ emission occurred within a high pressure ridge compared to emissions occurring west of this blocking condition.

Figure 1 shows the "typical" temporal development of $O_3$ and $CH_4$ due to an aviation attributed $NO_x$ emission for two emission locations next to each other (45°N 45°W and 45°N 30°W), representative for the examples presented in Grewe et al. (2017a) and Frömming et al. (2020). Here, one emission region is inside and the other one is west of a high pressure ridge (see top panel in Fig. 1). While the emitted $NO_x$ decreases in both air parcels, the $O_3$ concentration increases due to the described production processes (Reaction R1 - R3). Additionally, the emitted $NO_x$ leads to an elimination of $HO_x$ by forming nitric acid

**Table 1.** The integrated $O_3$ and $CH_4$ contribution to atmospheric concentration for both locations given in Fig. 1. The first location (45°N 45°W) is west of and the second location (45°N 30°W) within a high-pressure ridge. Additionally, the resulting climate impact for both locations, represented by Climate Change Functions (CCF), is given. For further details on the climate impact see Frömming et al. (2020). Integrals are given in kg days and CCF values are given in $K\,kg(N)^{-1}$.

| - | 45°N 45°W | 45°N 30°W |
|---|---|---|
| Integral $O_3$ | $469.6 \times 10^7$ | $600.3 \times 10^7$ |
| Integral $CH_4$ | $-99.9 \times 10^7$ | $-158.5 \times 10^7$ |
| CCF $O_3$ | $1.65 \times 10^{-12}$ | $2.31 \times 10^{-12}$ |
| CCF $CH_4$ | $-9.19 \times 10^{-13}$ | $-9.56 \times 10^{-13}$ |
| CCF PMO [a] | $-2.67 \times 10^{-13}$ | $-2.78 \times 10^{-13}$ |

[a] Primary mode ozone (PMO)

($HNO_3$) and peroxynitric acid ($HNO_4$):

$$OH + NO_2 + M \quad \rightarrow \quad HNO_3 + M \tag{R5}$$

$$HO_2 + NO_2 + M \quad \rightarrow \quad HNO_4 + M \tag{R6}$$

Both, $HNO_3$ and $HNO_4$ are subsequently removed by heterogeneous or other reactions, leading to an exponential decay of
$NO_x$ and a complete elimination after about a month. When the $NO_x$ mixing ratio is below a certain level, only little $O_3$ is produced and loss terms dominate the $O_3$ chemistry leading to a continuous decrease in the formed $O_3$. At the same time, the additional $O_3$ and $NO_x$ form OH which leads to a depletion of $CH_4$ (Reaction R4). After all $NO_x$ and $O_3$ is lost, the negative $CH_4$ anomaly starts to decay and will later reach its original values (not seen in Figure 1). For the two emission regions, the resulting $O_3$ gain differs, with the emission region in the high pressure system having an earlier $O_3$ maximum with a higher
magnitude. In contrast, the $CH_4$ depletion is only characterised by a varying magnitude. This large variability in the $NO_x$-$O_3$-$CH_4$ relation, induced by the same $NO_x$ emission, is also presented in Fig. 9 of Grewe et al. (2014b). However, the question remains if the different resulting characteristics for both emission locations given in Fig. 1 can be explained by the different weather conditions experienced by each air parcel.

     Within the present study, we investigate the impact of weather situations on changes in $O_3$ and $CH_4$ concentrations induced
by $NO_x$ emissions in the upper troposphere of the North Atlantic flight sector. In general, it is common to analyse the integrated $O_3$ change or the integrated radiative forcing induced by changes in $O_3$. Table 1 gives the integrated $O_3$ and $CH_4$ for both regions presented in Fig. 1. Additionally, the so-called Climate Change Functions (CCFs), a measure on how the Earth surface temperature will change due to a locally restricted $NO_x$ emission (for more details see Grewe et al. (2014b), Grewe et al. (2014a), and Frömming et al. (2020)), is given. The intention of this manuscript is not to identify correlations between weather
conditions and the resulting climate impact. Still, understanding the relation between the resulting climate impact and typical characteristics of the contribution to $O_3$ and $CH_4$ provides valuable insights. An earlier and larger $O_3$ contribution correlates with a higher integrated $O_3$ concentration and a higher resulting climate impact. Analysing the integrated $O_3$ and $CH_4$ is not

feasible when analysing the influence of weather conditions on the induced composition changes due to aviation attributed $NO_x$ emissions. Comparing varying weather conditions to a single data point (e.g. the integrated $O_3$) is difficult. For instance, a higher statistical significance is expected when analysing the mean value of a typical weather condition over a 20 day mean until the $O_3$ maximum is reached (for location 2 in Fig. 1 and Table 1) instead of the complete 90 days period, due to the chaotic nature of weather conditions. Thus, typical characteristics of the temporal development of $O_3$ and $CH_4$ are more suitable for this analysis since it is expected that they are directly influenced by varying weather conditions. Therefore, we especially focus on how weather conditions influence the time when the $O_3$ maximum occurs, the total $O_3$ gained, as well as the total $CH_4$ depleted. Our findings are additionally analysed with respect to inter-seasonal variability. This is achieved by using the results of simulations performed in the European project REACT4C (Reducing Emissions from Aviation by Changing Trajectories for the benefit of Climate, https://www.react4c.eu/ (Matthes, 2011)). The modelling approach of REACT4C as well as the methodology used in this study is elaborated in Section 2. Afterwards all findings of this study will be presented (Section 3). In Section 4, uncertainties and findings of this study will be discussed including a possible implementation strategy.

## 2 Methodology

Our analysis of the general concept of REACT4C as well as the modelling approach used will be elaborated first, to understand how the impacts of $NO_x$ emissions on $O_3$ and $CH_4$ were simulated. The idea of the project is presented by Matthes (2011) and Matthes et al. (2012). A complete description of the modelling approach used is given by Grewe et al. (2014b). Afterwards, a detailed description of the steps taken within the analysis of this work is presented.

### 2.1 REACT4C

REACT4C investigated the feasibility of adapting flight routes and flight altitudes to minimise the climate impact of aviation and to estimate the global effect of such air traffic management (ATM) measures (Grewe et al., 2014b). In this particular study, this mitigation option was tested over the North Atlantic region. The general steps in this modelling approach were as follows: (1) select representative weather patterns, (2) define time-regions, (3) model atmospheric contributions for additional emissions in these time-regions, (4) calculate the adjusted radiative forcing (RF), (5) calculate the climate change function (CCF) for each emission species and induced cloudiness, (6) optimize aircraft trajectories, based on the CCF results, by using an air traffic simulation system (System for traffic Assignment and Analysis at a Macroscopic level, SAAM), which is coupled to an emission tool (Advanced Emission Model, AEM), and (7) calculate the resulting operation costs and the resulting climate impact reduction. For the present study, only step one to three are important and will be further elaborated.

Irvine et al. (2013) identified that by simulating frequently occurring weather situations within a season, the global seasonal impact can be estimated. They analysed meteorological reanalysis data for 21 years for summer and winter. This reanalysis leads to three distinct summer (SP1-3) and five winter (WP1-5) patterns. The different weather patterns mainly vary in their location, orientation and strength of the jet stream and the phase of the North Atlantic Oscillation and the Arctic Oscillation. A graphical representation of each defined weather pattern is given by Irvine et al. (2013, Fig. 7 and Fig. 8 for winter and

summer, respectively) and the actual weather situations simulated in REACT4C are presented in Frömming et al. (2020). Due to the lower variability of the jet stream in summer, only three distinct weather situations were determined. The summer patterns occur 19 (SP1), 55 (SP2) and 18 (SP3) and the winter patterns 17 (WP1 & WP2), 15 (WP3 & WP4) and 26 (WP5) times per season in the reanalysis data (Irvine et al., 2013). Analogously, REACT4C simulated eight distinct model days, each representing one of these weather patterns.

To calculate the climate change functions, a time-region grid was defined in the North Atlantic region for seven latitudes (between 30°N to 80°N) and six longitudes (between 80°W to 0°W) over 4 different pressure levels (200, 250, 300 and 400 hPa) to account for different flight levels. At each time-region grid point, unit emissions of $CO_2$, $NO_x$ and $H_2O$ are initialised on 50 trajectories at 6, 12 and 18 UTC. However, Grewe et al. (2014b) found that the results show only minor sensitivity with respect to the temporal resolution. Therefore, only 12 UTC is considered in this study. The 50 trajectories are randomly located in the respective model grid box in which the specific time-region grid point is located. At each time-region grid point, $5 \times 10^5$ kg of NO (equals $2.33 \times 10^5$ kg(N)) is emitted, which is then equally distributed onto the trajectories (Grewe et al., 2014b).

## 2.2 Base model description

The ECHAM/MESSy Atmospheric Chemistry (EMAC) model is a numerical chemistry and climate simulation system that includes sub-models describing tropospheric and middle atmosphere processes and their interaction with oceans, land and human influences (Jöckel et al., 2010). It uses the second version of the Modular Earth Submodel System (MESSy2) to link multi-institutional computer codes. The core atmospheric model is the 5[th] generation European Centre Hamburg general circulation model (ECHAM5, Roeckner et al. (2003)). For the present study we applied EMAC (ECHAM5 version 5.3.02, MESSy version 2.52.0) in the T42L41-resolution, i.e. with a spherical truncation of T42 (corresponding to a quadratic Gaussian grid of approximately 2.8 by 2.8 degrees in latitude and longitude) with 41 vertical hybrid pressure levels up to 5 hPa.

The applied model setup comprised multiple MESSy submodules important for the performed simulations. Each of the tracers (i.e. $NO_x$ and $H_2O$) is emitted in an air parcel by the submodel TREXP (Tracer Release EXperiments from Point sources). The air parcel is then advected by the submodel ATTILA (Atmospheric Tracer Transport In a LAgrangian model) (Reithmeier and Sausen, 2002) using the wind field from EMAC. In addition to the 50 air parcels with tracer loading starting at each time-region, empty background air parcels are modelled in the northern hemisphere to allow for additional mixing, which in total yields about 169000 air parcels. The air parcels have a constant mass and the mixing ratio of each species is defined on the parcels' centroid. The centroid is assumed to be representative for the whole air parcel and the Lagrangian cells are considered isolated air parcels. While ATTILA is per se non-diffusive, inter-parcel mixing is parameterized by bringing the mass mixing ratio in a parcel closer to the average background mixing ratio, which is the average mixing ratio of all parcels within a grid box. The vertical transport due to subgrid-scale convection in ATTILA is calculated in three steps: (1) mapping the ATTILA tracer concentrations from the air parcels to the EMAC grid, (2) calculating the convective mass fluxes similarly as for standard EMAC tracers, and (3) mapping the calculated tendencies back to the air parcels. While a gain of tracer mass is

distributed evenly among the air parcels in a grid cell, a reduction of tracer mass is calculated according to the mass available. Further details are given in Reithmeier and Sausen (2002).

For each trajectory, the contribution of the emission (i.e. $NO_x$ and $H_2O$) to the atmospheric concentration of $CH_4$, $O_3$, $HNO_3$, $H_2O$, and OH is calculated over a time period of 90 days by using the submodel AIRTRAC (version 1.0, Frömming et al. (2013); see Supplement of Grewe et al. (2014b)). The tagging approach used by AIRTRAC was first described by Grewe et al. (2010). In this approach, each important chemical reaction is doubled. The first reaction applies to the whole atmosphere (from here onwards referred to as background) and the second one only to the additionally emitted tracer (from here onwards referred to as foreground). The submodel MECCA (Module Efficiently Calculating the Chemistry of the Atmosphere) is used to model the background chemical processes in the troposphere and stratosphere. The chemical mechanism used by MECCA can be grouped into sulfur, non-$CH_4$ hydrocarbon, basic $O_3$, $CH_4$, $HO_x$ and $NO_x$ and halogen chemistry (Sander et al., 2005). AIRTRAC on the other hand calculates the resulting changes due to the additional emitted $NO_x$ in the foreground. AIRTRAC assumes that each concentration change of $O_3$ due to aviation is attributed to the emitted $NO_x$, which is consistent with Brasseur et al. (1998). Concentration changes due to additionally emitted $NO_x$ are calculated based on the concentration of all chemical species involved in the general chemical system and the concentrations due to the extra emitted $NO_x$. The actual concentration change is then calculated based on the background reaction rate and the fraction of foreground and background concentrations of all reactants (Grewe et al., 2010). In detail, the foreground loss of $O_3$ ($L_{O_3}^f$) via Reaction R7 is based on the foreground and background concentrations of $NO_2$ and $O_3$ ($NO_2^f$, $O_3^f$ and $NO_2^b$, $O_3^b$ for foreground and background, respectively) and the background loss of $O_3$ ($L_{O_3}^b$), as given in Eq. (1).

$$NO_2 + O_3 \rightarrow NO + 2O_2 \tag{R7}$$

$$L_{O_3}^f = L_{O_3}^b \times \frac{1}{2}\left(\frac{NO_2^f}{NO_2^b} + \frac{O_3^f}{O_3^b}\right) \tag{1}$$

In total, AIRTRAC calculates the mass development of $NO_x$, $O_3$, $HNO_3$, OH, $HO_2$ and $H_2O$ by tracking 14 reactions and reaction groups. These can be split into: (1) one group for both the production and the destruction of $O_3$, (2) one reaction for the formation of $HNO_3$, (3) three and five reactions for the OH production and destruction, and (4) three reaction groups for the production and destruction of $HO_2$. Further, loss processes like wash-out and deposition are taken into account (Grewe et al., 2014b). The results of this mechanism agree well with earlier studies with respect to the regionally different chemical regimes and the overall effect of aviation emissions (Grewe et al., 2017c, Section 4.3, therein). The tagging mechanism also enables the quantification of $CH_4$ losses due to the two major reaction pathways, the change in $HO_x$ partitioning towards OH due to a $NO_x$ emission, and the production of OH due to an enhanced $O_3$ concentration (Grewe et al., 2017c, Figure 8). Section 4 includes an elaborate discussion on the modelling approach used.

## 2.3 Analysis performed in this study

Within this study we used the simulation output created by the REACT4C project. As some output variables were not available for all emission locations and weather patterns, not all time-regions and weather patterns could be included in the present study. Some of the raw data of WP2 were subject to data loss and not all analyses could be performed with this weather pattern.
Therefore, WP2 has been excluded from the analysis. From originally 1344 emission locations, 1115 are analysed. At each emission location all 50 air parcels are taken into account, resulting in 55750 trajectories being analysed. An output resolution of six hours was used over 90 days.

The variables taken into account can be categorised into three different groups: (1) background and foreground chemical concentrations, (2) background and foreground chemical reaction rates, and (3) general weather information. All foreground variables are present on the tracer grid, whereas background data are stored on the original EMAC grid. To simplify the analysis, background data were re-gridded onto the tracer grid. Here, it has been assumed that all background data within a grid box are valid for each air parcel within this specific EMAC grid box.

Due to the general complexity of the atmospheric chemistry, many variables can potentially influence changes in $O_3$ and $CH_4$ concentrations induced by $NO_x$ emissions. Therefore, correlation matrices were used to identify interacting parameters. For these matrices the three most common statistical measures to identify correlations were used (Pearson, Kendall, and Spearman's rank correlation coefficient). Statistical significance is ensured by using t-tests, one-way analysis of variance (ANOVA) and Tukey's honest significant difference (HSD) tests.

The long term reduction of $O_3$ due to the induced $CH_4$ loss (i.e. PMO) occurs far beyond the 90 days simulated by RE-ACT4C. Simulating the effect of PMO explicitly is computationally too expensive for the modelling approach used. In RE-ACT4C, PMO was thus not modelled explicitly. Instead a constant scaling factor of 0.29, based on Dahlmann (2012), was applied to the resulting climate impact of $CH_4$ (Grewe et al., 2014b). The effect of PMO is therefore not considered in this study and focus is only on the total $CH_4$ depletion.

## 3 Results

Within this section, the results of this study are described. First, a short analysis of the characteristics of the variability in the $O_3$ maximum is presented. The influence of transport processes on the time of the $O_3$ maximum is analysed in Section 3.2. The mechanisms controlling total $O_3$ gained are investigated in Sec. 3.3. The influence of tropospheric water vapour on total $CH_4$ depletion is discussed in Section 3.4. All findings are presented for summer and winter. An inter-seasonal variability analysis is performed in Section 3.5.

### 3.1 Characteristics of the temporal development of $O_3$

Figure 2 shows the maximum $O_3$ mixing ratio in relation to the time after emission when the maximum occurs. During winter and summer, high concentration changes are only possible if the $O_3$ maximum occurs early. In this scope, the $O_3$ maximum

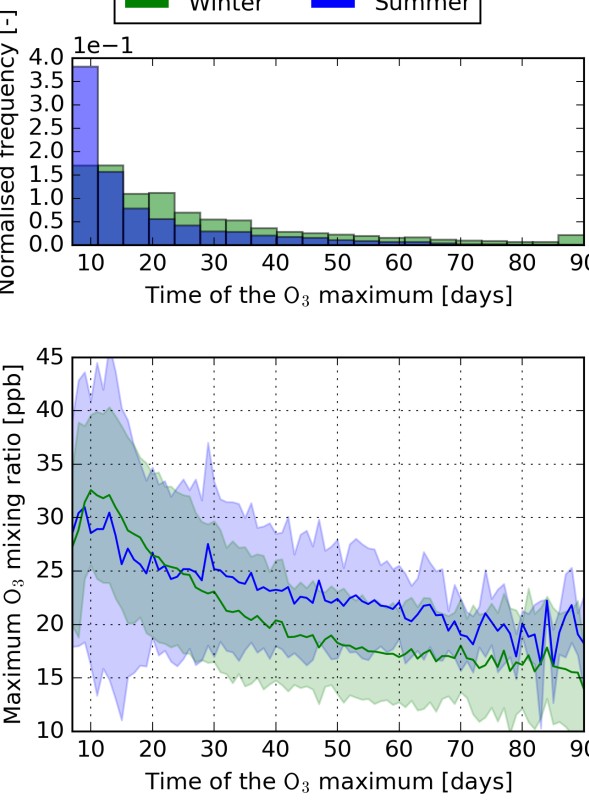

**Figure 2.** Top: Normalised histogram of frequencies when the $O_3$ maximum is reached after emission. Results are normalised to the total number of air parcels in summer and winter, respectively. Bottom: Mean (solid line) and standard deviation (shaded area) of the maximum $O_3$ mixing ratio in relation to the time when the $O_3$ maximum is reached.

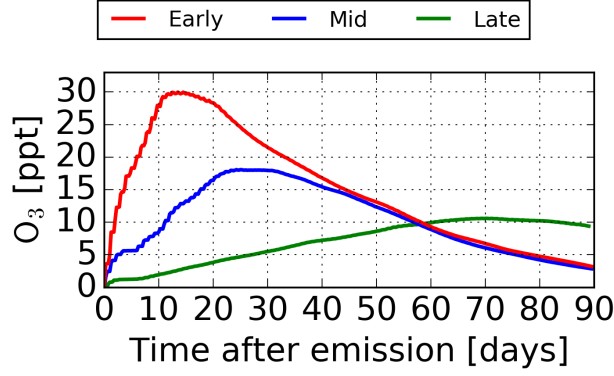

**Figure 3.** "Typical" temporal development of an early (red), a mid (blue) and a late (green) $O_3$ maximum.

is defined as the maximum mixing ratio after which no further increase of $O_3$ occurs. Figure 3 shows the "typical" temporal development of $O_3$ for an early, a mid, and a late $O_3$ maximum. The early maximum is characterised by a high production of $O_3$ in the first days after emission. The mid and late maximum are dominated by a slower $O_3$ production. In the case of the late maxima, the extreme slow $O_3$ production leads to a stretched version of the temporal development of the air parcels that have an early or a mid maximum. As stated in Sec. 1, only the early maximum is characterised by a high $O_3$ maximum and the magnitude decreases by 1/3 and 2/3 for the mid and late maxima, respectively. The top panel of Fig. 2 gives the frequency of when the maximum occurs for both seasons. About 47.5 % and 72 % of all air parcels reach their $O_3$ maximum during the first 21 days in winter and summer, respectively. Only a small number of air parcels do not have defined maxima within the 90 days of simulation (winter: 2.5 %, summer: < 1 %). At the end of the simulation period, the $O_3$ concentration of these air parcels is still increasing. However, almost all $NO_x$ is removed at the end of simulation. It is thus expected that the formed $O_3$ would be quickly reduced, if the simulation was continued beyond the 90 days of simulation. All of these air parcels are emitted at higher altitudes (200 or 250 hPa) and higher latitudes (more than 70 % are emitted north of 50°N). During winter, these air parcels are transported to latitudes north of 70°N and do not experience any solar radiation (i.e. polar night). The missing solar radiation dampens the $O_3$ formation, leading to no distinct $O_3$ maximum within the 90 simulated days.

## 3.2  Importance of transport processes on the time of the $O_3$ maxima

It is well established that the $O_3$ production efficiency depends on the general chemical activity, controlled by weather conditions (i.e. temperature), and the concentration of each reactant. These weather conditions and reactant concentrations differ significantly across the troposphere, such that certain regions have a higher $O_3$ production efficiency. Therefore, the transport into these regions controls the $O_3$ gained. Our analysis shows that air parcels with an early maximum are characterised by a strong downward wind component, whereas late maxima have a weak downward or even an upward vertical wind component (not shown). Therefore, air parcels with an early $O_3$ maximum are those that are transported to lower altitudes (top panel in Fig. 4) and lower latitudes (bottom of Fig. 4). Air parcels with a late maximum mostly stay at the emission altitude and latitude or are transported to higher altitudes and latitudes. For all winter patterns, most maxima occur in a region spanning from 15°N to 35°N at pressure altitudes between 900 to 600 hPa. The maximum region is slightly shifted to higher altitudes for all summer patterns. No maximum occurs at high latitudes during winter due to the absence of solar radiation in the polar region. This indicates that a significant $O_3$ production, leading to an early $O_3$ maximum, is only possible if an air parcel is transported to lower altitudes and latitudes.

Tropospheric vertical transport processes have many causes, e.g. temperature differences, incoming solar radiation, as well as latent and sensible heat fluxes. Vertical transport occurs in conveyor belt events and causes an exchange of trace gases between the upper and the lower troposphere. Figure 5 (top) shows the mean 250 hPa geopotential height anomaly. The geopotential height is an approximation of the actual height of a pressure surface (here 250 hPa) above the mean sea-level. Here, the anomaly is presented for a better comparison of summer and winter, due to generally higher geopotential heights during summer. The anomaly is obtained by deducting the seasonal mean. In classical weather analysis, the geopotential height is used to identify synoptic weather systems. The positive deviation in the geopotential height for air parcels with an early maximum indicates

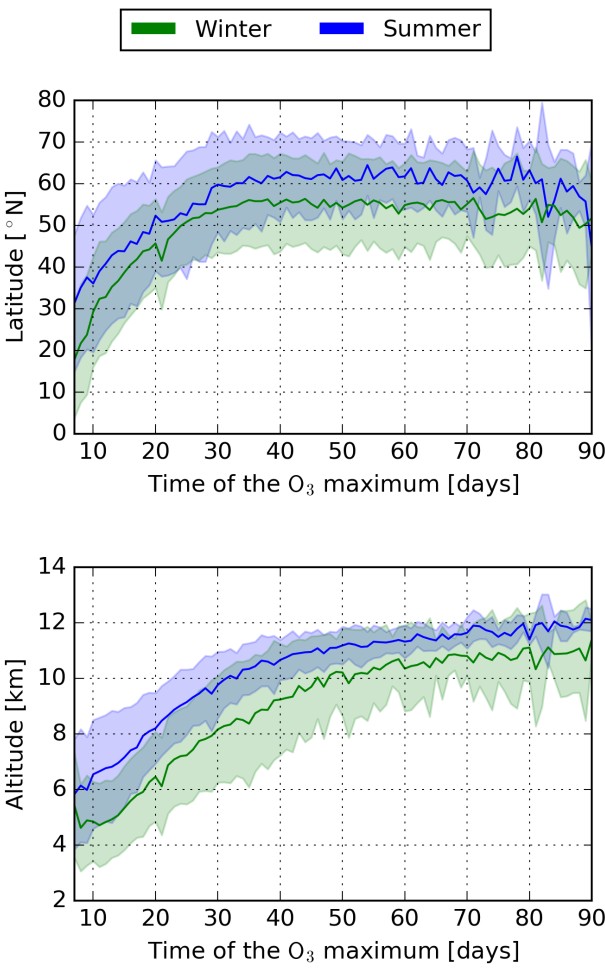

**Figure 4.** Mean (solid line) and standard deviation (shaded area) of the location of each air parcel within the first seven days after emission in relation to when the maximum $O_3$ mixing ratio is reached. Top: Latitude. Bottom: Altitude.

that these air parcels originate or are transported into and stay within a high pressure system. Air parcels originating from the core of a high pressure system have generally earlier maxima compared to air parcels which are transported into high pressure systems after emission (not shown). It is well known that subsidence is dominating vertical transport processes within high pressure systems, explaining the strong downward motion of air parcels, characterised by early maxima. Air parcels with late maxima stay within low pressure systems where upward motion dominates.

### 3.3 Weather conditions controlling the $O_3$ production efficiency

Even though early $O_3$ maxima are characterised by strong vertical downward transport, transport processes do not directly influence chemical processes in the atmosphere. Temperature is known to be a major factor controlling chemical processes

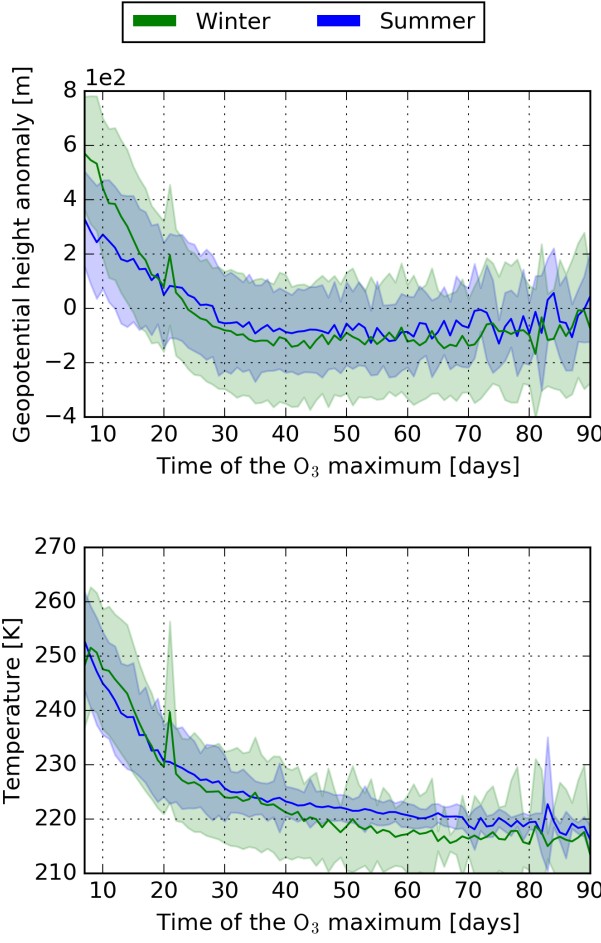

**Figure 5.** Top: Mean 250 hPa geopotential height anomaly. The mean is calculated for the first seven days after emission, similar to the mean values given in Fig. 4. The anomaly is calculated based on the seasonal mean. Bottom: Mean dry air temperature. Here, the mean is calculated based on the time between emission and the $O_3$ maximum. Both parameters are given in relation to the time of the $O_3$ maximum. The seasonal mean is represented by the solid line and the standard deviation as shaded area.

in the atmosphere and is generally higher at lower altitudes and latitudes. The bottom panel of Fig. 5 shows the mean dry air temperature along the air parcel trajectory until the $O_3$ maximum is reached. The mean dry air temperature is higher for air parcels with early $O_3$ maxima, which is due to the downward and southward transport (leading to higher temperatures) within high pressure systems. These higher temperatures lead to higher background chemical activity (higher background reaction rates) and therefore accelerate foreground chemistry. Higher temperatures and enhanced photochemical activity at higher altitudes during Northern Hemispheric (NH) summer, explain the tendency of earlier maxima in this season.

From classical chemistry the efficient production of $O_3$ does not only depend on high chemical activity due to higher temperatures, but also on the concentrations of the reactants involved. In the case of the formation of $O_3$ due to $NO_x$, these are

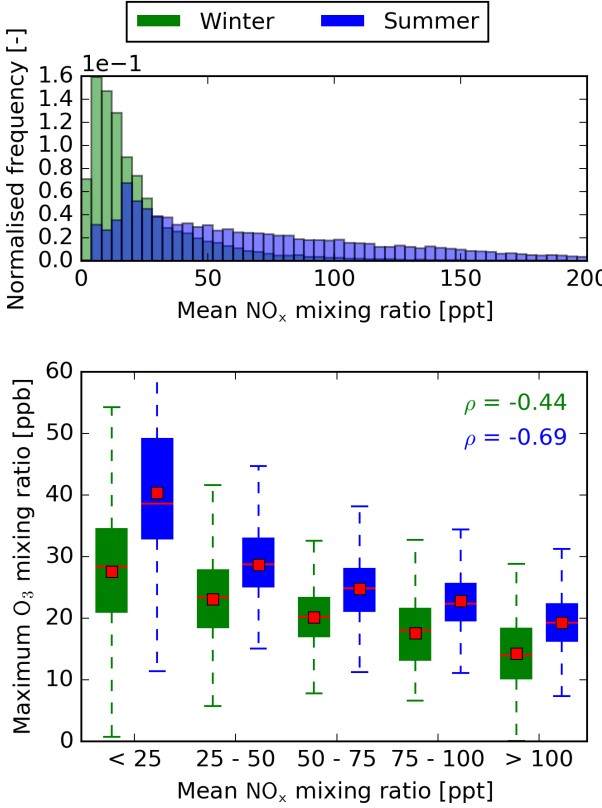

**Figure 6.** Top: Normalised histogram of the mean $NO_x$ mixing ratio until the $O_3$ maximum is reached. Results are normalised to the total number of air parcels in summer and winter, respectively. Bottom: Binned box plots showing the relation between the mean $NO_x$ mixing ratio and the magnitude of the $O_3$ maximum. The median and mean for each box plot are represented by red lines and red boxes, respectively. Additionally, the Spearman rank coefficient is given for summer and winter.

NO and $HO_2$ (Reaction R1). Figure 6 and 7 show how the mixing ratios of $NO_x$ and $HO_x$, respectively, relate to the maximum $O_3$ mixing ratio. Here, $NO_x$ and $HO_x$ are used to account for the rapid cycling of the species within each radical group. For both seasons, only low $NO_x$ concentrations will lead to high $O_3$ contributions. The production of $O_3$ via Reactions R1 to R3 dominates at low background $NO_x$ concentrations, whereas at high background concentrations, $NO_2$ is eliminated by reacting

5 with OH and $HO_2$ forming $HNO_3$ and $HNO_4$, respectively. From Fig. 7 it becomes evident that a high increase in $O_3$ is only possible at high $HO_x$ concentrations, since at low $HO_2$, no $O_3$ will be formed via Reaction R1.

In the upper troposphere, the background $NO_x$ concentration is altitude dependent and generally increases towards the tropopause (not shown). On the other hand, $HO_x$ is high at low altitudes and latitudes and decreases towards the tropopause. Air parcels with a fast downward transport generally experience lower mean background $NO_x$ concentrations and higher $HO_x$

10 background concentrations, resulting in a higher $O_3$ gain. Air parcels, which stay close to the tropopause or are even transported into the stratosphere, experience a high background $NO_x$ concentration, leading to a lower $O_3$ formation since most $NO_x$ is

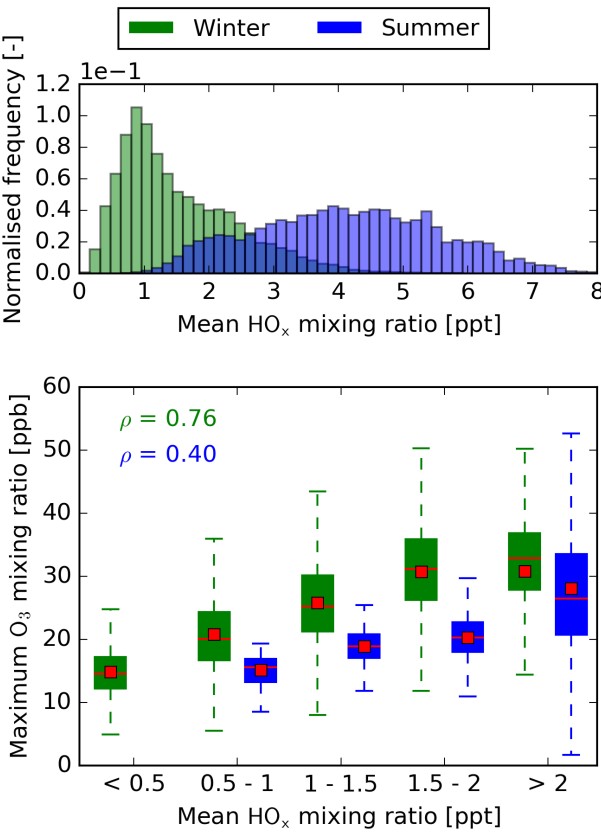

**Figure 7.** Top: Normalised histogram of the mean $HO_x$ mixing ratio until the $O_3$ maximum is reached. Results are normalised to the total number of air parcels in summer and winter, respectively. Bottom: Binned box plots showing the relation between the mean $HO_x$ mixing ratio and the magnitude of the $O_3$ maximum. Please note that during summer, $HO_x$ mixing ratios below 0.5 ppt show no statistical significance. Therefore, no box plot is provided in this case. The median and mean for each box plot are represented by red lines and red boxes, respectively. Additionally, the Spearman rank coefficient is given for summer and winter.

eliminated by forming $HNO_3$ and $HNO_4$. In addition, high background $HO_x$ concentrations dominate in high pressure systems (not shown) explaining why air parcels in high pressure systems have a generally higher $O_3$ formation.

Summer and winter significantly differ with regard to the correlation of $NO_x$ and $HO_x$ with the $O_3$ maximum, respectively. Summer has a high Spearman rank coefficient for $NO_x$ ($\rho = -0.69$) but only correlates weakly with $HO_x$ ($\rho = 0.40$). Winter on the other hand correlates well with $HO_x$ ($\rho = 0.76$) but weakly with $NO_x$ ($\rho = -0.44$). This difference is explained by varying $NO_x$ and $HO_x$ concentrations in both seasons. The top panels of Fig. 6 and 7 give the normalised frequencies of $NO_x$ and $HO_x$ for both seasons. It becomes evident that winter is characterised by low $NO_x$ and $HO_x$ concentrations, where as summer is dominated by high $NO_x$ and $HO_x$ concentrations. For many air parcels during summer, enough $HO_x$ is available to allow a high formation of $O_3$, but a higher $NO_x$ concentration limits the formation of $O_3$ and leads to the formation of $HNO_3$ and

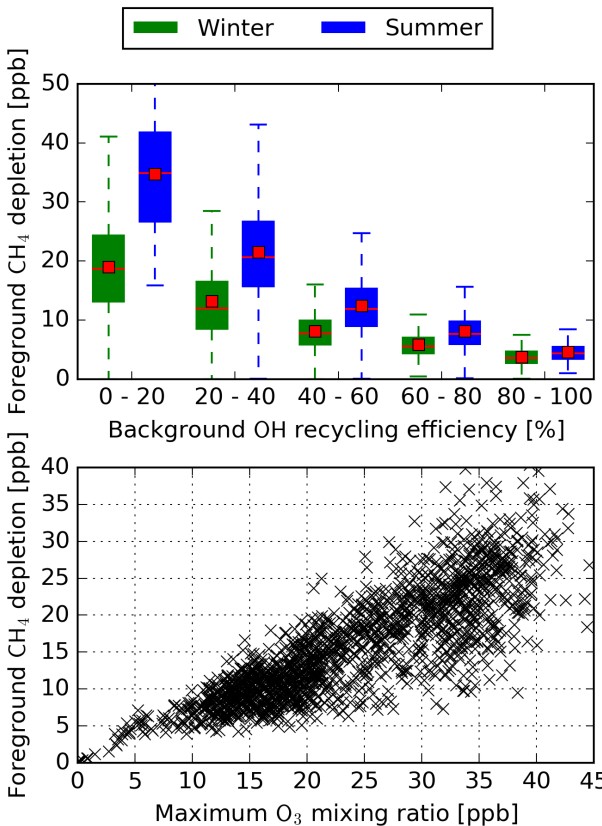

**Figure 8.** Top: Binned box plots of the OH recycling probability. The median and mean for each box plot are represented by red lines and red boxes, respectively. Bottom: Maximum $O_3$ mixing ratio vs. maximum $CH_4$ mixing ratio for recycling probabilities below 20 % for WP1.

$HNO_4$. During winter, the low $NO_x$ concentrations theoretically allow for a high $O_3$ formation, but low $HO_x$ concentrations limit the efficient production of $O_3$.

### 3.4 Influence of water vapour on the total $CH_4$ depletion

In comparison to $O_3$, only the total $CH_4$ depletion is of interest due to its longer atmospheric lifetime. A high $O_3$ concentration leads to a high $CH_4$ depletion (Spearman rank coefficient of 0.66), since $O_3$ is a major source of OH accelerating the depletion of $CH_4$ (Reaction R4). However, the moderate Spearman rank coefficient indicates that other factors additionally control the $CH_4$ depletion process. Our results show that a high foreground $CH_4$ depletion is only possible if the background OH concentration is high. When looking at OH, the fast cycling between OH and $HO_2$ has to be taken into account. Analysing the recycling probability (r) of OH is useful to account for this cycling. Here, we define the recycling probability, following Lelieveld et al. (2002), as:

$$r = 1 - \frac{P}{G} \tag{2}$$

in which P is the primary production of OH via:

$$H_2O + O(^1D) \rightarrow 2OH \tag{R8}$$

and G is the gross OH production additionally considering the secondary production of OH. In general, when r approaches 100 % the formation of OH becomes autocatalytic. When r approaches 0 %, all OH is formed via Reaction R8. Based on a

perturbation study, Lelieveld et al. (2002) identified that for recycling probabilities above 60 %, the chemical system becomes buffered and that $NO_x$ perturbations in this regime have only little impact on OH. The top of Fig. 8 shows that a high $CH_4$ depletion is only possible if the recycling probability of background OH is below 60 %. When the major source of OH is from Reaction R8 (r approaches 0 %), the formed OH is not recycled via $NO_x$, accelerating the depletion of $CH_4$. The major source of foreground OH is the formed $O_3$. A high background OH recycling probability does not necessarily mean that foreground

$O_3$ is efficiently produced. The foreground formation of $O_3$ is limited by background $NO_x$ and $HO_x$ during summer and winter, respectively. Figure 8 (bottom) shows the relation between the total $CH_4$ depletion and the $O_3$ magnitude for recycling probabilities below 20 % (most left box-plot in top of Fig. 8). Thus, the possible low $O_3$ formed limits the $CH_4$ depletion at low OH recycling probabilities, explaining the high spread in the total $CH_4$ depletion in that regime. It can thus be concluded that a high depletion of $CH_4$ is only possible if the major formation of OH is due to Reaction R8.

Globally, Lelieveld et al. (2016) estimate that about 30% of the tropospheric OH is produced by Reaction R8. This reaction is limited by the availability of $O(^1D)$, formed from the photolysis of $O_3$ and $H_2O$. In this study, the highest $CH_4$ depletion rate occurs in tropical regions close to the surface (between 0°and 20°N and below 850hPa), which are dominated by hot and humid weather conditions. These regions are known to have high $HO_x$ concentrations due to active photochemistry and large OH sources and sinks. Here, the contribution of OH being produced by water vapour is highest (Lelieveld et al., 2016). A

strong correlation exists between the average specific humidity along the air parcel trajectory and the total depletion of $CH_4$ (mean Spearman rank coefficient of 0.77). In particular, air parcels with a low OH recycling probability and thus a high $CH_4$ depletion are characterised by high specific humidity and high incoming solar radiation. Therefore, a high depletion of $CH_4$ is only possible if the air parcel is transported towards low altitudes and latitudes. The transport into tropical regions occurs mainly due to the subsidence in high pressure systems (see Sec. 3.2).

## 3.5  Inter-seasonal variability

Within this study, specific weather situations (for graphical representations see Irvine et al. (2013, their Fig. 7 and Fig. 8), and Frömming et al. (2020)) were analysed. Table 2 shows an overview of all correlations analysed within this study for each distinct weather pattern. Vertical transport processes until the $O_3$ maximum, represented by the vertical wind velocity, correlate reasonably well within both seasons. However, during summer the correlation tends to be lower. SP 3 has the lowest

correlation coefficient and the highest mean downward wind velocity. The pattern is characterised by a high pressure blocking situation, resulting in an overall high layer thickness, resulting in a weaker correlation. It is thus expected that differences in each individual weather situation (i.e. number, location and strength of the high pressure systems) cause the inter-seasonal variability. Additionally, downward transport during summer is less important for air parcels to experience high temperatures.

**Table 2.** Spearman rank coefficients of all identify relations for each individual weather pattern. All correlation factors related to the time and the maximum $O_3$ mixing ratio are calculated based on mean values for the time span between emission and the time of the $O_3$ maximum. The correlation between the total $CH_4$ depletion and specific humidity is based on the mean between time of emission and the time when the total $CH_4$ depletion is reached.

| Correlation factors | | Winter | | | | Summer | | |
| --- | --- | --- | --- | --- | --- | --- | --- | --- |
| | | WP1 | WP3 | WP4 | WP5 | SP1 | SP2 | SP3 |
| Time of $O_3$ maximum | Vertical wind velocity | -0.67 | -0.77 | -0.69 | -0.61 | -0.66 | -0.66 | -0.57 |
| | 250 hPa geopotential height | -0.71 | -0.66 | -0.61 | -0.63 | -0.57 | -0.61 | -0.55 |
| | Mean temperature | -0.78 | -0.84 | -0.80 | -0.79 | -0.86 | -0.82 | -0.81 |
| Maximum $O_3$ mixing ratio | Mean background $NO_x$ | -0.08 | -0.37 | -0.43 | -0.66 | -0.73 | -0.76 | -0.78 |
| | Mean background $HO_2$ | 0.76 | 0.71 | 0.71 | 0.38 | 0.38 | 0.21 | 0.40 |
| Total $CH_4$ depletion | Specific humidity | 0.81 | 0.82 | 0.82 | 0.75 | 0.74 | 0.71 | 0.72 |

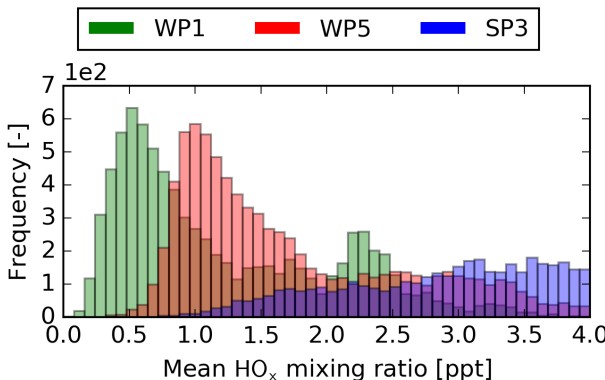

**Figure 9.** Histogram for the mean $HO_x$ mixing ratio until the $O_3$ maximum is reached for WP1, WP5, and SP3. Note that mixing ratios above 4 ppt are not shown.

This also explains the weaker correlation for the 250 hPa geopotential height during summer. Still, air parcels which stay in a high pressure region experience earlier maxima during summer.

The mean background concentration of $NO_x$ correlates strongly with the $O_3$ magnitude for each SP, giving no indication that there is another parameter that controls the total $O_3$ gain. However, no correlation exists for most WPs. This indicates that in winter, when chemistry is slow at emission locations at mid and higher latitudes, the transport pathway, e.g. towards the tropics, is more important than the chemical background conditions at the time of emission, whereas in summer with active photochemistry, the background $NO_x$ concentration plays a dominant role. Only WP5 shows a stronger correlation. At the same time, WP5 has a low correlation with $HO_x$. Figure 9 gives the frequencies of the mean background $HO_x$ concentrations

for WP1, WP5 and SP3. In comparison to WP1, WP5 is characterised by higher $HO_x$ concentrations with most mixing ratios above 1 ppt (WP1 mean: 1.2 ppt, WP5 mean: 1.8 ppt; WP1 median: 0.89 ppt, WP5 median: 1.4 ppt). In general, mixing ratios above 1 ppt allow a high $O_3$ formation (see Fig. 7). This indicates that for most air parcels in WP5, $HO_x$ is not the only limiting factor for the $O_3$ gain. At the same time WP5 is characterised by higher $NO_x$ mixing ratios (WP1 mean: 18.1 ppt,

WP5 mean: 31.6 ppt) explaining the higher correlation with $NO_x$. This indicates that in the case of WP5 the maximum $O_3$ concentration is limited by a combination of $HO_x$ and $NO_x$. The chosen example day in EMAC for WP5 occurs at the end of February, whereas the other WPs are initialised in December or early January. This indicates that $HO_x$ becomes a less limiting factor towards spring. This suggests that our results are only valid for both analysed seasons and further research is necessary to identify the controlling factors in spring and autumn.

Specific humidity is clearly the controlling factor of the total $CH_4$ depletion for all weather patterns taken into account. Again the correlation is weaker in summer, which is due to the generally higher $H_2O$ concentrations. This results in a lower variability in the specific humidity, which weakens the correlation analysed. Here, WP5 again behaves like all SPs. This indicates that $O_3$ and $CH_4$ concentration changes due to emissions in spring are most likely controlled by mechanisms identified for summer.

## 4   Uncertainties and discussion

Our results indicate a large impact of transport patterns on $O_3$ and $CH_4$ concentration changes due to aviation $NO_x$ emissions. This is both a highly complex interaction of transport and chemistry, and a relatively small contribution of $O_3$ and $CH_4$ concentration changes against a large natural variability. Hence, a direct validation of our results is not feasible. However, the main processes, such as transport and chemistry can be evaluated individually – at least in parts. In the following paragraphs, we will discuss some aspects of this interaction and the ability of EMAC to reproduce observations.

An important aspect in our study is the model's transport. Short-lived species, which only have a surface source such as $^{222}$Radon ($^{222}$Rn, radioactive decay half-lifetime of 3.8 d), are frequently used to validate fast vertical transport characteristics. Jöckel et al. (2010) and in more detail Brinkop and Jöckel (2019) showed that the model is able to capture the $^{222}$Rn surface concentrations and vertical profiles, indicating that the vertical transport is well represented in EMAC.

The horizontal transport is difficult to evaluate and observed trace gases, which resemble the exchange between mid and
high latitudes and the tropics, are not available. However, Orbe et al. (2018) compared transport timescales in various global models, e.g. from northern mid-latitudes to the tropics, which differed by 30%. The interhemispheric transport differed by 20%. The authors concluded that vertical transport is a major source of this variability. More research is needed to better constrain models with respect to their tropospheric transport timescales. A more integrated view on the variability of aviation related transport-chemistry interaction is given by a model intercomparison of $NO_x$ concentration differences between a simulation
with and without aircraft $NO_x$ emissions (Søvde et al., 2014, see their supplementary material). When concentrating only on their winter results to reduce the chemical impact to a minimum, the results clearly show a very similar $NO_x$ change of the 5 models (including EMAC) peaking around 40°N at cruise altitude in winter with a tendency to a downward and southward transport to the tropics. However, the peak values vary between 55 pptv and 70 pptv.

Chemistry, or more specific, the concentrations of chemically active species are evaluated in detail in Jöckel et al. (2010) and Jöckel et al. (2016). In general, EMAC overestimates the tropospheric $O_3$ column by 5-10 DU in mid-latitudes and 10-15 DU in the tropics. Carbon monoxide on the other hand is underestimated, though the variability matches well with observations. The tropospheric oxidation capacity is at the lower end of model estimates, but within the models' uncertainty ranges. Jöckel et al. (2016) speculate that lightning $NO_x$ emissions or stratosphere-to-troposphere exchange might play a role. It is important to note that variations, which are caused by meteorology are in most cases well represented (e.g. Grewe et al., 2017a, their Sec. 3.2).

Ehhalt and Rohrer (1995) already stated that the net-$O_3$ gain strongly depends in a non-linear manner on the $NO_x$ mixing ratio. This is generally well reproduced in EMAC (Mertens et al., 2018, Figure 5) and even by an EMAC predecessor model (Dahlmann et al., 2011; Grewe et al., 2012, their Figure 4 and Figure 1, respectively). Stevenson et al. (2004) showed the response of $O_3$ and $CH_4$ to a pulse $NO_x$ emission, which is very similar to our results (Grewe et al., 2014b, their Figure 9). Stevenson and Derwent (2009) demonstrated that the $NO_x$ concentration at time of emissions strongly defines the resulting climate impact of $O_3$. A similar but weaker relation can be found in the current data set (not shown) (Grewe et al., 2014b, their Figure 9). For some air parcels the background $NO_x$ concentration is low at time of emission, but they are quickly transported to regions characterised by higher concentrations. These air parcels experience a temporal high $O_3$ production shortly after emission due to low $NO_x$ concentrations, but only little total $O_3$ is formed due to a high mean background $NO_x$ mixing ratios after emission. This explains the difference between our correlation and the one of Stevenson and Derwent (2009). Another aspect is the question how well the REACT4C concept, to model atmospheric effects of a local emission (Grewe et al., 2014b), represents global modelling approaches, such as Søvde et al. (2014) or Grewe et al. (2017c). The approach was developed to gain more insights in aviation effects that can conventionally not be obtained. Hence, there are by definition limitations in answering this question. However, four indications can be given, which support the consistency of the modelling approach. First, the transport scheme is reasonably well established (Brinkop and Jöckel, 2019, and above). Second, the chemical response to a local emission agrees well with earlier findings of Stevenson et al. (2004), who simulated the monthly mean response of $O_3$ and $CH_4$ to a $NO_x$ pulse and which are very similar to the results from this approach (Grewe et al., 2014b, their Fig. 9). Third, the use of a trajectory analysis to interpret either observational data or modelling data is well established (Riede et al., 2009; Cooper et al., 2010). And fourth, a first verification of the resulting global pattern of the atmospheric sensitivity to a local $NO_x$ emission by comparing to sparsely available literature data was promising (Yin et al., 2018). This verification is based on a generalisation of the CCFs by developing algorithms to relate the weather information available at the time of emission to the resulting CCF. These algorithmic CCFs (aCCF, van Manen and Grewe (2019)) allow to predict all weather situations, compared to the limited applicability to a few selected days for the CCFs. An annual climatology of the results from using aCCFs was calculated and compared to conventional approaches by Yin et al. (2018). They conclude: It "shows the variation pattern of the ozone aCCFs matches well with the literature results over the northern hemisphere (the latitude between 30° N and 90°N) and the flight corridor (roughly 9 km to 12 km vertical range)".

From our findings, it can be concluded that not only the atmospheric conditions at the time of emission influences the $O_3$ gain but rather the region in which the maximum $O_3$ concentration and the maximum concentration change occur. The findings

of Stevenson and Derwent (2009) are also only valid for summer and no winter analysis is provided, making it impossible to directly compare our findings identified in Sec. 3.3 to available literature. However, indirectly, by the use of generalised aCCFs, first findings indicate a reasonably well agreement of the simulated atmospheric response to local $NO_x$ emissions.

To conclude, both transport and chemistry processes are crucial for our results. EMAC is in many aspects in line with other model results, but has some biases in the concentration of chemical species. However, the variability of chemical species, such as $NO_x$ and $O_3$ is better represented than mean values, indicating that the interaction between transport and chemistry is reasonably well simulated. This result should be robust, since our results show a very strong relation between meteorology and the contribution of aviation emission, and since this interaction is in principle well represented in EMAC. However, the strength of the $O_3$ response to a $NO_x$ emission in a high pressure system has an uncertainty, which we hardly can estimate. Based on the results of the model intercomparison by Søvde et al. (2014), we would expect an uncertainty in the order of 25%.

The globally increasing aviation activity and the resulting increase in the contribution of aviation to anthropogenic climate change (Lee et al., 2009) result in the necessity to find possible mitigation strategies (Matthes et al., 2012). One possible mitigation strategy is to re-route flights based on their potential climate impact. The feasibility of this concept was demonstrated by Grewe et al. (2014a, 2017b). However, mitigating the climate impact from aviation by estimating the climate impact and re-route flight trajectories using the same simulation setup (in resolution, time horizon and chemical mechanism used) as in REACT4C on a day-to-day basis is currently computationally too expensive and not feasible at the moment. van Manen and Grewe (2019) and Yin et al. (2018) demonstrated that the CCFs defined in REACT4C can be approximated by algorithms based on meteorological parameters on the day of emission, which significantly reduce the computational demand. The main findings of the present study are a step towards a better understanding of the influence of weather conditions. It allows to suggest an alternative approach, which is in-between the detailed CCFs of REACT4C and the aCCFs suggested by van Manen and Grewe (2019). All factors identified in the present study provide a correlation to the resulting climate impact but can not be used as a new climate metric. Still, algorithmic CCFs could be defined to estimate the resulting climate impact. This would allow to use computationally cheaper entirely dynamic simulations. Two possible weather factors to approximate the resulting climate impact are: (1) vertical wind velocity for the impact of $O_3$, and (2) specific humidity for the depletion of $CH_4$. Table 3 gives the Spearman correlation of both factors in dependency of the simulation time taken into account. It becomes obvious that the correlation is enhanced with the number of days taken into account. Thus, even shorter dynamical simulations would be sufficient for this approximation. The next step would be to develop aCCFs based on the first days after emission. If implemented into general forecasting services, these aCCFs could be used to re-route aviation on a day-to-day basis with low computational demand. However, investigating the feasibility of this approach is beyond the scope of this manuscript.

## 5 Conclusions

The possibility to reduce aviation's climate impact by avoiding climate sensitive regions, heavily depends on our understanding of the driving influences on induced contributions to the chemical composition of the atmosphere. In this study, we demonstrated the importance of transport processes on locally induced aviation attributed $NO_x$ emission on $O_3$ and $CH_4$ concentra-

**Table 3.** Spearman rank coefficients for the correlation between the vertical transport and the time of the $O_3$ maximum, and the specific humidity and the total $CH_4$ loss. Spearman rank coefficients are provided in depends of the period used to calculate the mean value. The Spearman rank coefficient for each correlation for the period from the $NO_x$ emission until the $O_3$ maximum is also given.

| Correlation | At emission | 2 days | 3 days | 4 days | 5 days | 6 days | 7 days | Till $O_3$ max. |
|---|---|---|---|---|---|---|---|---|
| Mean vertical transport vs. time of $O_3$ maximum | -0.19 | -0.30 | -0.39 | -0.47 | -0.51 | -0.57 | -0.68 | -0.69 |
| Mean specific humidity vs. total $CH_4$ loss | 0.48 | 0.50 | 0.52 | 0.54 | 0.55 | 0.59 | 0.61 | 0.78 |

tions over the North Atlantic flight sector. The induced $O_3$ change is characterised by the time and magnitude of its maximum and high $O_3$ maxima are only found if the maximum occurs early. Transport processes like subsidence in high pressure systems lead to early maxima due to the fast transport into regions with a higher chemical activity. In summer, the $NO_x$-$HO_x$-relation is limited by background $NO_x$, whereas in winter, the limiting factor are low $HO_x$ concentrations. When an air parcel is trans-
ported into regions with high $NO_x$ concentrations in summer, a low change in total $O_3$ occurs, since less $O_3$ is formed in the background. In this case, most $NO_2$ is eliminated by forming $HNO_3$ and $HNO_4$. During winter, low background $NO_x$ concentrations allow for a high $O_3$ formation, but due to generally lower $HO_x$ concentrations no efficient $O_3$ formation occurs. Air parcels transported quickly towards lower altitudes encounter low $NO_x$ but high $HO_x$ concentrations leading to a higher $O_3$ formation, strengthening the importance of transport processes on the $O_3$ formation.
The total depletion of $CH_4$ depends heavily on the background OH concentration. If most OH is formed by its primary formation process, which depends on water vapour, and only little OH is recycled to $HO_2$, a high depletion of $CH_4$ occurs. Therefore, the water vapour content, which the air parcel experiences along its trajectory, defines the total $CH_4$ depletion. Air parcels transported into lower altitudes and latitudes experience higher water vapour concentrations. Thus, atmospheric transport processes also define the total $CH_4$ depletion. Additionally, only high total $O_3$ gains correlate with a large $CH_4$
depletion.
    Due to the complexity of the problem, we are not able to validate our results. It would be challenging to design a measurement campaign to proof the contribution of aviation $NO_x$ emissions to $O_3$ and $CH_4$. The standard deviation of background concentrations are generally considered to be higher than changes induced by aviation $NO_x$ perturbation, making them hardly detectable (Wauben et al., 1997). Our analysis of the model performance however shows that both transport processes as well
as chemical concentrations are reasonably well represented. Our inter-seasonal analysis shows that our findings to the importance of background $NO_x$ and $HO_x$ concentrations are only valid for both seasons analysed. Due to the high variability of $NO_x$ and $HO_x$ concentrations in the troposphere, we expect other factors to control the total $O_3$ gain in other regions, not analysed in this study. Based on the findings of Köhler et al. (2013), we expect our results to be valid for most parts of the northern

extra-tropics. To conclude, further model studies are necessary to fully quantify how transport processes influence induced changes of $O_3$ and $CH_4$ concentrations in all seasons as well as other regions of interest.

Our results show that transport processes are of most interest when identifying the impact of local $NO_x$ emissions on $O_3$ and $CH_4$. Since, entirely dynamic simulations without chemistry are computationally less expensive, the insights gained in this work suggest a more feasible approach where the climate impact would be estimated based on transport processes and other weather factors within the first days of simulation. Short term dynamic simulations would reduce the computational demand and would thus make re-routing flights on a day-to-day basis possible.

*Data availability.* The data of the REACT4C project used in this work are archived at the German Climate Computing Centre (Deutsches Klimarechenzentrum, DKRZ) and are available on request.

*Author contributions.* SR and VG designed the analysis and SR carried it out. CF performed the simulations of REACT4C. SR prepared the manuscript with contributions from all co-authors.

*Competing interests.* The authors declare that they have no conflict of interest.

*Acknowledgements.* This work was supported by the European Union FP7 Project REACT4C (Reducing Emissions from Aviation by Changing Trajectories for the benefit of Climate: www.react4c.eu/, Grant Agreement Number 233772) and contributes to the DLR project Eco2Fly. Computational resources were made available by the German Climate Computing Center (DKRZ) through support from the German Federal Ministry of Education and Research (BMBF) and by the Leibniz-Rechenzentrum (LRZ). We would like to thank Mariano Mertens from DLR for providing an internal review.

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
