# Peer review of "The impact of weather pattern and related transport processes on aviation's contribution to ozone and methane concentrations from $NO_x$ emissions"

_Atmospheric Chemistry and Physics, 2020_

## Referee Comment (RC1) · William Collins (Referee) · 26 Feb 2020

This study addresses how the impact of aircraft NOx emissions on ozone and methane vary according to the meteorological conditions. The concept is useful, but a lot more work is needed to present a coherent analysis.

The key variable used seems to be the time of the ozone maximum, but the explanation of why this is chosen is hard to discover. The most obvious variables to use would be

the integrated ozone perturbation (i.e. the area under the curve in figure 1). Or the integrated radiative forcing (i.e. integral of O3 scaled by a radiative efficiency as a function of altitude and latitude). For instance figure 2 shows a correlation between the maximum O3 concentration and time of the O3 maximum, but it is not at all clear that one is actually driving the other. Presumably both these are simply identifying regions of high O3 production efficiency.

It is not at all clear even what the time of the O3 maximum means at longer timescales. Presumably in these cases the O3 timeseries doesn't look like a stretched version of figure 1, but rather a varying timeline that happens to bump up at 40 or 50 days for some meteorological reason. Any NOx signal will have long dissipated after 20 days (figure 1) so it is not obvious that there is any physical meaning to the later ozone maxima. Example timeseries for "early" and "late" maxima need to be shown.

Nearly all the analysis is done for the winter (figs 2, 3, 4, 6, 7) when the effects will be far smaller than in the summer. The magnitude of the winter and summer impacts need to be compared. There is no need to consider the winter at all if it turns out to be unimportant, and certainly the analysis should focus on the summer.

Line 12: It is not at all obvious that the time of maximum should be the controlling factor, rather than the magnitude of the maximum.

Line 13: It is more likely that the subsidence leads to greater ozone production efficiency, and that the earlier ozone maximum is a consequence of this, rather than a cause of it.

Line 15: This seems to be stating the obvious – the size of the CH4 decrease depends only on the size of the CH4 decrease.

Line 29: Presumably the aim of this study is to identify those meteorological conditions that are conducive to ozone formation so that the computationally expensive chemical

trajectories are not needed?

Line 30: It is not explicitly stated that the aim is to avoid producing ozone, but to enhance the destruction of methane.

Line 12: It needs to be made clear in these sentences whether the climate impact is warming or cooling. It would help to contrast the effects on ozone and methane.

Line 4: The method for calculating the trajectories needs to be described. How do they account for sub-grid scale vertical motion?

Figure 1: Is this figure a change in the global burden? It would be useful to show changes along a trajectory since that is what is used in all subsequent figures.

Figure 2: How well is the "Time of the O3 maximum" defined? It might be that after 20 days there is no well-defined peak, but rather fluctuations of greater or lesser magnitude.

Line 5: Why is it assumed that the early maximum is the cause? It could just as easily be written that an early O3 maximum is only possible if the concentration change is high.

Line 7: The processes involved here need to be understood. It could be that higher altitude emissions don't produce much ozone, so that any fluctuations in ozone appear as spurious "late" maxima. The timeseries for these late maxima need to be shown.

Line 9-13: The RF or CCF is not mentioned again in this study. It appears they come from other work with the REACT4C project. Unless these can be related to the case

studies analysed here it is not helpful to discuss them. For example the comments on Lacis et al. (1990) refer to a higher radiative efficiency at altitude in contrast to the lower ozone production efficiency found in this study. Why do the time and magnitude of the O3 maximum influence the climate impact? Instead it seems it should be the integral of the ozone perturbation with a radiative efficiency factor for latitude and altitude. What is CCF and how is it determined?

Figure 3: The caption says the analysis is based on the first seven days after emission, but the figures show values out to 90 days.

Line 2-3: It is not obvious why the altitude difference is the crucial variable, rather than the absolute altitude. The discussion makes plausible arguments about increased ozone production at lower altitudes, therefore it would seems more logical to plot the altitudes where the trajectory ends up (maybe the mean altitude in the first 20 days), whereas there doesn't seem to be any argument that it is the amount of descent that is important. Except obviously that if the emissions all occur at similar flight levels then greater descent will give lower trajectories.

Line 5-6: The use of the time of maximum as the controlling variable is not obvious. For instance the claim that the earlier maxima in summer give less time for downward transport is much more likely to be due to the enhanced photochemistry in the summer giving more ozone production at higher altitudes, hence for a descending trajectory the maximum will occur earlier.

Line 8: Rather than focussing on the early ozone maximum, it would be more scientifically rigorous to state that significant ozone production only occurs if an air parcel is transported to lower altitudes and latitudes.

[Figure]

Line 2: To what extent is sub-grid scale convection included in the trajectory calculations?

Lines 14-24: In section 3 the argument was that earlier O3 maxima lead to greater ozone production. But here the opposite argument is being made – that increased ozone destruction leads to earlier maxima. In which case the early O3 maxima should be associated with less ozone not more. This is another example of why the time of the O3 maxima should not be used as a controlling variable. Note also that while reactions R2 to R4 might have negative temperature dependencies the origin of the HO2 and RO2 has strong positive temperature dependence, so higher temperatures do lead to more ozone production.

Line 2: This sentence doesn't seem correct. Do you mean to correlate NOx with ozone maxima?

Lines 10-14: This study uses prescribed emissions of NOx (5x10ˆ5 kg) so there doesn't seem any value in comparing this to NOx concentrations in an aircraft study. I suggest removing this paragraph.

Line 32: Stevenson and Derwent only analysed summer as ozone production and methane depletion are not important in winter.

Page 16:

Line 14: There has been no calculation of "resulting climate impact" in this study, therefore it is not clear how this can be a conclusion from this work.

---

## Referee Comment (RC2) · Anonymous Referee #2 · 2 Mar 2020

This work aims to provide insight into the effect that aviation NOx has on the climate through its contribution to tropospheric ozone and depletion of atmospheric methane. In particular, the authors aim to show how the specific weather pattern into which the emission occurs could affect the long-term chemical impact. They evaluate this using a trajectory-tracking approach with simplified tropospheric chemistry, embedded within a global chemistry-climate simulation. Using the time and magnitude of the maximum change in ozone as proxies for the total ozone-related climate impact, they find the overall radiative forcing is likely to change significantly depending on the season and

local meteorological conditions.

The central question of the paper is both interesting and important. If robust, this work could be a significant advance in the field of understanding the interactions between aviation and the atmosphere. It could also provide valuable, actionable information on how to reduce aviation's climate impacts. However, I have concerns regarding the methodological approach, and regarding the metric used to quantify "climate impact".

As such, I believe that major revisions are needed before this paper is ready for publication in ACP. I list major concerns below first, followed by minor issues.

Major comments

While innovative, it is not clear to me that the chemistry and physics simulated within the "tracked" air parcel will provide an estimate of aviation's impacts on the atmosphere which are accurate or consistent with the "parent" EMAC model.

1. Is it validated against any estimates using more conventional techniques? The authors cite Grewe et al (2017c) Section 4.3, but this does not appear to have an explicit comparison of the Lagrangian model's result for aviation impacts compared to other studies. A direct comparison showing (e.g.) the ozone and methane response in EMAC when performing a conventional simulation of an increment in aviation NOx would be very helpful. Alternatively, if such a comparison is already present in (eg) Grewe et al (2017c), a quantitative evaluation (e.g. "agreement to within X% when simulating aviation emissions") would be helpful. If not already present in Grewe et al 2017c I recommend that such an analysis be added to Section 5.

2. More detail on the Lagrangian model would be helpful. For example, what is the total air mass of the well-mixed box? How is diffusion treated? This is important because of the role of non-linear chemistry (see e.g. Kraabøl et al 2002), which could result in suppressed ozone production when concentrations are very high (i.e. early in the plume's development).

[Figure]

The approach used appears to quantify the direct effect that aviation NOx and H2O could have on the climate through increases in short-term ozone and decreases in methane, on the basis that both are greenhouse gases. However, one of the major effects of aviation NOx is a long-term reduction in tropospheric ozone, driven by the methane loss (see e.g. Holmes et al 2001). Is this accounted for here? If I have understood the trajectory-tracking mechanism correctly it does not include feedbacks, so I would not expect it to be accounted for. If so, it would be useful to include an estimate of the total magnitude of the missing long-term ozone loss.

The analysis is predicated on the idea that the time and magnitude of the peak change in ozone is a significant indicator of overall climate impacts. However, it is not clear to me that this is the case, and I could not find a quantitative justification or citation to this effect in the paper. The closest I found was the assertion that, in REACT4C, a higher ozone concentration change "generally" leads to a larger RF (p7, line 12). Why use these metrics instead of (e.g.) the total integrated ozone perturbation over the time of the simulation?

At the end of section 2.1, it is stated that "50 trajectories [are initialized] at 6, 12 and 18 UTC", but this is then immediately followed by "in the present study, only 12 UTC is considered". I'm confused – why have the first statement? Also, what is the error which we can expect from only including one time point? Won't this result in same geographical biases?

Many of the conclusions seem to treat correlations as causal links (e.g. section 4.3). Some conclusions – such as that "during summer the O3 formation is limited by the background NOx concentration, whereas in winter low HO2 concentrations limit the total O3 gained" – do not seem to be sufficiently supported by the data. It would be helpful to see a more explicit justification for why this must be the mechanism. More generally, much of the analysis is not very quantitative – such as page 14, lines 3-5 which states that a visual inspection of a correlation makes it "evident" that winter pattern 5 looks "almost the same" as for summer pattern 3. I strongly recommend that the

authors redo this analysis in a more quantitative fashion and remove conclusions which cannot be both quantified and mechanistically explained in a way which is supported by data.

Some of the analysis used for chemistry is difficult to interpret, and seems to ignore the cycling nature of certain key species. For example, on page 13, it is stated that "35% and 42% of background OH is produced by $HO_2$ reacting with $O_3$ and NO, respectively". However, OH and $HO_2$ are expected to be cycling rapidly. As such, "production" of OH in this fashion is hard to interpret, since it usually matters more to consider what the sources of HOx are. I am therefore skeptical of the claim that $H_2O$ is only a minor source of OH. I recommend that the authors rephrase this discussion to be about the OH/$HO_2$ ratio and the production and loss of HOx than to talk separately about OH and $HO_2$.

Minor issues

P2 l22: I went back and checked the claim that Gilmore et al (2013) showed that "during summer the climate impact is up to 1.5 times higher and only half in winter, when compared to the annual meaning". I do not think this is true. They did show that the ozone production efficiency was 50% higher than the average in summer, but this is largely compensated by changes in ozone lifetime. The overall change in ozone production rate is only about 10% above the annual mean in summer, and correspondingly about 10% below the mean in winter (see Figure 1 of said paper).

The claim that "the standard deviation of background concentrations [of ozone and NOx] are generally considered to be higher than changes induced by aviation. . ..making them hardly detectable" (p16 l30, and paraphrased on page 14, lines 17-19) is based on a single, outdated study. Wauben et al 1997 uses a 1995 aircraft NOx inventory, now 20-25 years old. Since total aviation NOx emissions have likely more than doubled since then (e.g. Wasiuk et al 2016), I think this claim either needs a more recent and robust backing or it should be removed.

The manuscript has several spelling and grammar mistakes. For example, in several locations (e.g. p2, l34) the authors use the word "adopt" when I suspect "adapt" is intended, and on p12 line 19 the word "exited" should be "excited". I would recommend another sweep through the manuscript to fix these and other typos.

What happened to WP2 in Table 1?

References

Holmes, C. D., Tang, Q. and Prather, M. J.: Uncertainties in climate assessment for the case of aviation NO, Proc. Natl. Acad. Sci. U. S. A., 108(27), 10997–11002, 2011.

Kraabøl, A. G., Berntsen, T. K., Sundet, J. K. and Stordal, F.: Impacts of NOx emissions from subsonic aircraft in a global three-dimensional chemistry transport model including plume processes, J. Geophys. Res. D: Atmos., 107(D22), ACH–22, 2002.

Wasiuk, D. K., Khan, M. A. H., Shallcross, D. E. and Lowenberg, M. H.: A Commercial Aircraft Fuel Burn and Emissions Inventory for 2005–2011, Atmosphere , 7(6), 78, 2016.

---

## Author Comment (AC1) · 18 Jun 2020

Reply to the Review by William Collins (Referee)

Thank you very much for the helpful comments. We revised the text and Figures significantly and took all comments into account. Please find in black the original comments from William Collins and in red our reply.

This study addresses how the impact of aircraft $NO_x$ emissions on ozone and methane vary according to the meteorological conditions. The concept is useful, but a lot more work is needed to present a coherent analysis.

Thank you very much for your review. Your input helped to improve the manuscript significantly. In order to provide a higher consistency within this work, we adapted major parts of the manuscript. Please find below an overview on the changes, based on the referees' comments, which will be discussed in more detail further below:

1. Abstract: The abstract was rephrased to better represent the work and results of this work
2. The introduction was extended to cover the chemical processes and an elaborated discussion why the $O_3$ maximum (w.r.t. time and magnitude) is selected for the analysis
3. The model description now includes a description of the lagrangain model used
4. Section 3 was eliminated since the chemical system is now covered in the introduction. A subsection was added to the results, discussing the variability of the $O_3$ maximum and how typical temporal developments of early, mid and late $O_3$ maxima look like.
5. The altitude analysis was changed to now focus on the mean altitude and not the altitude difference
6. The influence of temperature, $NO_x$ and $HO_x$ in summer and winter are now moved to a single section discussing the $O_3$ production efficiency
7. In order to account for the rapid cycling between OH and $HO_2$, we now analyze the OH recycling probability when discussing the $CH_4$ depletion
8. The discussion now includes a section on how the lagrangain modelling approach influences the results
9. All figures were updated to show the relation for winter and summer and make them visually more appealing

The key variable used seems to be the time of the ozone maximum, but the explanation of why this is chosen is hard to discover. The most obvious variables to use would be the integrated ozone perturbation (i.e. the area under the curve in figure 1). Or the integrated radiative forcing (i.e. integral of $O_3$ scaled by a radiative efficiency as a function of altitude and latitude). For instance figure 2 shows a correlation between the maximum $O_3$ concentration and time of the $O_3$ maximum, but it is not at all clear that one is actually

driving the other. Presumably both these are simply identifying regions of high $O_3$ production efficiency.

In general we agree that the most obvious variable for addressing the climate impact is the integrated ozone perturbation. And this is the starting point of our investigation. The integrated values differ for emission locations. The aim of our paper is to understand these differences in more detail. Hence, identifying the resulting climate impact is not the target of this paper. In this work we are mainly interested in how transport processes affect the resulting ozone and methane change. By using our dataset, we identify that two characteristics (time and magnitude of the ozone maximum) differ most under varying weather conditions. We added an additional figure (Figure 1 in the revised version), a table (Table 1 in the revised version) and an additional paragraph to the introduction clarifying the purpose of this manuscript. In the figure we show that two emission regions next to each other lead to different ozone perturbations, characterized by a different time and magnitude. Since, the emission occurs under different weather conditions (in and west of a high pressure ridge), we open the question if the differences with regard to the ozone perturbations can be explained by the different weather conditions experienced by the air parcels.

With the revised Figure 2 we want to demonstrate that only if the ozone maximum occurs early a high ozone magnitude is possible but that also early maxima might have only a low maximum. Within our manuscript we demonstrate that early maxima are only possible if the air parcel is transported towards lower altitudes and therefore into regions with a higher chemical activity. The magnitude of the ozone maximum still depends if the air parcel is transported into regions favoring high ozone production efficiency.

It is not at all clear even what the time of the $O_3$ maximum means at longer timescales. Presumably in these cases the $O_3$ time series doesn't look like a stretched version of figure 1, but rather a varying timeline that happens to bump up at 40 or 50 days for some meteorological reason. Any $NO_x$ signal will have long dissipated after 20 days(figure 1) so it is not obvious that there is any physical meaning to the later ozone maxima. Example time series for "early" and "late" maxima need to be shown.

In the case of late $O_3$ maxima the temporal development looks like a stretched version of early maxima. The following figure illustrates the typical development of an early, a mid and a late $O_3$ maximum:

[Figure]

The corresponding NOx concentration looks as follows:

[Figure]

In the case of the late $O_3$ maximum (green line) $NO_x$ is first reduced mainly due to the formation of $HNO_3$ (due to high background $NO_x$ concentrations). After 20 days the air parcel is transported towards high latitudes with a largely reduced solar radiation (close to polar night). Due to the reduced solar radiation, only little $O_3$ is formed over time, leading to a low and late $O_3$ maximum.

We added the first figure to the revised manuscript (Figure 3) including a discussion on these characteristics.

Nearly all the analysis is done for the winter (figs 2, 3, 4, 6, 7) when the effects will be far smaller than in the summer. The magnitude of the winter and summer impacts need to be compared. There is no need to consider the winter at all if it turns out to be unimportant, and certainly the analysis should focus on the summer.

In a companion paper (submitted to ACPD, acp-2020-529) we present the resulting climate impact of all weather situations. In general, the climate impact is indeed higher during summer, however not negligible in winter. In the following figure (a similar figure can be found in the companion paper (their Figure 8)), we show the trajectories of two air parcels released at higher latitudes during winter (marked A and B):

[Figure]

Both air parcels are released close to each other, but only air parcel A is transported towards lower altitudes and into the tropics, leading to a higher climate impact. Obviously the air parcels are released at different location within a weather situation leading to different transport pathways. Hence even if the chemistry is slow at the location of the emission, the transport of the emitted species to tropical regions leads to a significant chemical processing. Therefore there is a clear difference between the seasons. Thus, the ozone contribution from $NO_x$ emissions in winter is not negligible.

The following figure from the companion paper (their Figure 13) gives the climate impact due to changes in ozone and methane for winter (blue) and summer (red):

[Figure]

It becomes evident that the resulting climate impact during winter has a similar variability and therefore similar or even higher re-routing possibilities. Thus, winter can be considered equally interesting for our purpose.

We still see the necessity to show also results for summer and thus modified all figures to show the results and correlations for both seasons.

The variability of the ozone response with respect to the emission location and season has been shown in Frömming et al, the companion paper. Here we are more interested in understanding the differences.

The following five comments are all related to the abstract. After receiving your review, it became obvious that the abstract does not represent the intention and the findings of this manuscript adequately. Thanks a lot for the helpful comments! We therefore decided to rephrase major parts of the abstract.

Line 12: It is not at all obvious that the time of maximum should be the controlling factor, rather than the magnitude of the maximum.

You are indeed correct, since the time of the maximum is not the controlling factor. However, only early maxima allow for a high total ozone change. We rephrased this part and also included what we identify as the two major characteristics of the ozone perturbation (time and magnitude of the ozone maximum). See also the discussion above.

Line 13: It is more likely that the subsidence leads to greater ozone production efficiency, and that the earlier ozone maximum is a consequence of this, rather than a cause of it.

In this study we find that the subsidence leads to transport into regions of higher chemical activity. This higher chemical activity then leads to the earlier maximum. We rephrased this section to explain this relation and to better represent the findings of this work.

Line 15: This seems to be stating the obvious – the size of the $CH_4$ decrease depends only on the size of the $CH_4$ decrease.

True. We removed this section.

Line 29: Presumably the aim of this study is to identify those meteorological conditions that are conducive to ozone formation so that the computationally expensive chemical trajectories are not needed?

Ultimately yes. This paper tries to be a step towards an improved understanding of the relation between the actual weather situation at the time of emission and the ozone response. We rephrased parts of the abstract and the introduction, to make this more obvious for the reader.

Line 30: It is not explicitly stated that the aim is to avoid producing ozone, but to enhance the destruction of methane.

The re-routing approach presented in Grewe et al. 2014a and Grewe et al. 2014b performs the optimization based the total climate impact from $NO_x$. Regions with a high overall climate impact are avoided and regions with a low climate impact are favored, independent

if the lower climate impact is caused by less ozone being produced or more methane being depleted. You are raising an important discussion, which has been addressed in another paper: Grewe et al. 2017:

"In our approach, the routes which reduce the climate impact avoid regions where warming contrails are formed or the ozone impact is large. However, routes are also favored, where contrails contribute to cooling or the emission of $NO_x$ leads to a methane reduction which cools more than the increase in ozone warms. This raises the question, to what extent should additional contrail formation be allowed, which—over a chosen time span—cools the global climate more than the additional $CO_2$ emitted by climate optimized routing warms. These questions have to be considered carefully for any climate-optimized routing."

We think this paper is not the right forum to further discuss this, here we are focusing on the understanding of the atmospheric processes, whereas any application of climate-optimized routing must address these issues. We do not think that there is a clear scientific answer to the question, it might be more a decision by policy makers and society guided by science.

Grewe, V., Frömming, C., Matthes, S., Brinkop, S., Ponater, M., Dietmüller, S., Jöckel, P., Garny, H., Tsati, E., Dahlmann, K., Søvde, O. A., Fuglestvedt, J., Berntsen, T. K., Shine, K. P., Irvine, E. A., Champougny, T., and Hullah, P.: Aircraft routing with minimal climate impact: the REACT4C climate cost function modelling approach (V1.0), Geosci. Model Dev., 7, 175–201, https://doi.org/10.5194/gmd-7-175-2014, 2014a.

Grewe, V.; Champougny, T.; Matthes, S.; Frömming, C.; Brinkop, S.; Sø vde, A.; Irvine, E.; Halscheidt, L.Reduction of the air traffic's contribution to climate change: A REACT4C case study. Atmos. Environ. 94, 616–625, 2014b.

Grewe, V., Matthes, S., Frömming, C., Brinkop, S., Jöckel, P., Gierens, K., Champougny, T., Fuglestvedt, J., Haslerud, A., Irvine, E., Shine, K., Climate-optimized air traffic routing for trans-Atlantic flights. Environm. Res. Lett. 12(3), 034003, DOI: 10.1088/1748-9326/aa5ba0, 2017.

Line 12: It needs to be made clear in these sentences whether the climate impact is warming or cooling. It would help to contrast the effects on ozone and methane.

These are crucial information for the reader. We added two sentences covering this information.

Line 4: The method for calculating the trajectories needs to be described. How do they account for sub-grid scale vertical motion?

We added an elaborate explanation of the submodel (ATTILA) which is used to model the transport of each air parcel to the base model description. The sub-grid scale vertical motion in ATTILA is calculated in three steps. First, mapping the ATTILA tracer concentrations from the air parcels to the EMAC grid. Second, calculating the convective mass fluxes similarly as for standard EMAC tracers. Third, mapping the calculated tendencies back to the air parcels.

Figure 1: Is this figure a change in the global burden? It would be useful to show changes along a trajectory since that is what is used in all subsequent figures.

In this figure the change in the global burden was shown. However, this figure has been removed in favor of the new figure added in the introduction (discussed earlier). This figure covers the same information for different positions. To be able to compare the results to the resulting climate impact the change in the global burden needs to be shown. However, figure 3 shows an early, a mid and a late $O_3$ maximum along the trajectories.

Figure 2: How well is the "Time of the $O_3$ maximum" defined? It might be that after 20 days there is no well-defined peak, but rather fluctuations of greater or lesser magnitude.

We defined the $O_3$ maximum such that it is the highest $O_3$ concentration with no further increase of $O_3$ afterwards. Information about this was added to the manuscript.

Line 5: Why is it assumed that the early maximum is the cause? It could just as easily be written that an early $O_3$ maximum is only possible if the concentration change is high.

You are right that the earlier $O_3$ maximum is only possible if the $O_3$ production is high. In this study we do not assume that the earlier maximum is the cause for the higher O3 maximum. Instead we just state that high $O_3$ maxima are possible if the $O_3$ maximum is reached early. This is of course only possible if the production efficiency is high (see Section 3.3).

Line 7: The processes involved here need to be understood. It could be that higher altitude emissions don't produce much ozone, so that any fluctuations in ozone appear as spurious "late" maxima. The time series for these late maxima need to be shown.

Please see our earlier answer on this matter. A graphic showing representative examples for early, mid and late maxima was added to the manuscript.

Line 9-13: The RF or CCF is not mentioned again in this study. It appears they come from other work with the REACT4C project. Unless these can be related to the case studies

analysed here it is not helpful to discuss them. For example the comments on Lacis et al. (1990) refer to a higher radiative efficiency at altitude in contrast to the lower ozone production efficiency found in this study. Why do the time and magnitude of the $O_3$ maximum influence the climate impact? Instead it seems it should be the integral of the ozone perturbation with a radiative efficiency factor for latitude and altitude. What is CCF and how is it determined?

The RF and CCF are indeed from other works within REACT4C. An elaborate explanation on how the CCFs are calculated, including a validation/verification, is presented in Grewe et al. (2014) and a more detailed analysis of the CCFs is given in more detail in our companion manuscript (acp-2020-529).  Within this study, we are concentrating on a specific aspect of the CCFs and we found that early and high $O_3$ maxima relate well to high RF and CCF values. The introduction was extended, to increase the understanding of this relation.

Grewe, V., Frömming, C., Matthes, S., Brinkop, S., Ponater, M., Dietmüller, S., Jöckel, P., Garny, H., Tsati, E., Dahlmann, K., Søvde, O. A., Fuglestvedt, J., Berntsen, T. K., Shine, K. P., Irvine, E. A., Champougny, T., and Hullah, P.: Aircraft routing with minimal climate impact: the REACT4C climate cost function modelling approach (V1.0), Geosci. Model Dev., 7, 175–201, https://doi.org/10.5194/gmd-7-175-2014, 2014.

Figure 3: The caption says the analysis is based on the first seven days after emission, but the figures show values out to 90 days.

The Figure 3 bottom shows the mean latitude of the air parcel for the first seven days after emission in relation to the time when the ozone maximum occur. We rephrased this caption to make clearer that the seven days are only related to the mean latitude and not the time of the ozone maximum. We used seven days for the latitudinal mean since the first ozone maxima occur after seven days. Using only seven days reduces the potential bias of air parcels with a late maximum.

Line 2-3: It is not obvious why the altitude difference is the crucial variable, rather than the absolute altitude. The discussion makes plausible arguments about increased ozone production at lower altitudes, therefore it would seems more logical to plot the altitudes where the trajectory ends up (maybe the mean altitude in the first 20 days),whereas there doesn't seem to be any argument that it is the amount of descent that is important. Except obviously that if the emissions all occur at similar flight levels then greater descent will give lower trajectories.

We agree that analyzing the absolute altitude makes more sense in the scope of this analysis. Therefore, the analysis was changed accordingly. However, instead of 20 days we only use the first seven days for calculating the mean altitude (see earlier explanation).

Line 5-6: The use of the time of maximum as the controlling variable is not obvious. For instance the claim that the earlier maxima in summer give less time for downward transport is much more likely to be due to the enhanced photochemistry in the summer giving more ozone production at higher altitudes, hence for a descending trajectory the maximum will occur earlier.

Agreed, the time of maximum is not the controlling factor, but an important piece of information in understanding the variations in the ozone response to a $NO_x$ emission (see also discussion above). Thank you very much for pointing this out. We checked the data and we find the same relation and added a section covering this relation to the analysis.

Line 8: Rather than focussing on the early ozone maximum, it would be more scientifically rigorous to state that significant ozone production only occurs if an air parcel is transported to lower altitudes and latitudes.

We agree on this. The text has been changed accordingly. In addition, we restructured the original Section 4.2 and 4.3 and merged it into a single Section 3.3 focusing on the $O_3$ production efficiency.

Line 2: To what extent is sub-grid scale convection included in the trajectory calculations?

We addressed this in an earlier answer above.

Lines 14-24: In section 3 the argument was that earlier $O_3$ maxima lead to greater ozone production. But here the opposite argument is being made – that increased ozone destruction leads to earlier maxima. In which case the early $O_3$ maxima should be associated with less ozone not more. This is another example of why the time of the $O_3$ maxima should not be used as a controlling variable. Note also that while reactions R2 to R4 might have negative temperature dependencies the origin of the $HO_2$ and$RO_2$ has strong positive temperature dependence, so higher temperatures do lead to more ozone production.

Thank you for pointing this out. The temperature relations are of course correct as you described them. With the new structure of Section 3.3 (focusing on $O_3$ production efficiency instead of temperature, $NO_x$ and $HO_x$ separately) this part is no longer necessary and was thus removed. In this study, the time of the $O_3$ maximum is not the controlling factor of the resulting climate impact but rather a typical characteristic of the temporal development of $O_3$. In general, the production phase is more interesting for this study, since the ultimate goal is to have short simulations to predict the climate impact from aviation and allow for

efficient re-routing. Using the 90 days integral is thus not feasible. An elaborate discussion on this manner was added to the introduction.

Line 2: This sentence doesn't seem correct. Do you mean to correlate $NO_x$ with ozone maxima?

We understand this confusion - apologies. In this sentence we intended to refer to the correlation between the mean $NO_x$ concentration and the magnitude of the ozone maximum. We updated the description of the relation in the new section accordingly.

Lines 10-14: This study uses prescribed emissions of $NO_x$ (5x10ˆ5 kg) so there doesn't seem any value in comparing this to $NO_x$ concentrations in an aircraft study. I suggest removing this paragraph.

A direct comparison of our results to other studies is complicated, since our approach was not used by any other modelling study yet. However, in Søvde et al. 2014 EMAC (our base model used) was compared to other models focusing on the impact of aviation emissions. In this part of the discussion, we focus on how well EMAC compares to other models and find that it compares reasonably well. We think that this is information is important and thus decided to keep it within our discussion.

Søvde, O. A., Matthes, S., Skowron, A., Iachetti, D., Lim, L., Owen, B., Øivind Hodnebrog, Genova, G. D., Pitari, G., Lee, D. S., Myhre,G., and Isaksen, I. S.: Aircraft emission mitigation by changing route altitude: A multi-model estimate of aircraftNOxemission impactonO3photochemistry, Atmospheric Environment, 95, 468 – 479, https://doi.org/https://doi.org/10.1016/j.atmosenv.2014.06.049, 2014

Line 32: Stevenson and Derwent only analysed summer as ozone production and methane depletion are not important in winter.

This is correct. However, analyzing the importance of weather conditions in winter is crucial, due to the high variability in the resulting climate impact and the possibility that air parcel are transported to equatorial regions during winter (see earlier discussion).

Page 16:

Line 14: There has been no calculation of "resulting climate impact" in this study, therefore it is not clear how this can be a conclusion from this work.

That is indeed correct. We changed the manuscript accordingly.

---

## Author Comment (AC2) · 18 Jun 2020

Reply to the review by the Anonymous Referee #2

Thank you very much for the helpful comments. We revised the text and figures significantly and took all comments into account. Please find in black the original comments from reviewer #2 and in red our reply.

This work aims to provide insight into the effect that aviation $NO_x$ has on the climate through its contribution to tropospheric ozone and depletion of atmospheric methane. In particular, the authors aim to show how the specific weather pattern into which the emission occurs could affect the long-term chemical impact. They evaluate this using a trajectory-tracking approach with simplified tropospheric chemistry, embedded within a global chemistry-climate simulation. Using the time and magnitude of the maximum change in ozone as proxies for the total ozone-related climate impact, they find the overall radiative forcing is likely to change significantly depending on the season and local meteorological conditions.

The central question of the paper is both interesting and important. If robust, this work could be a significant advance in the field of understanding the interactions between aviation and the atmosphere. It could also provide valuable, actionable information on how to reduce aviation's climate impacts. However, I have concerns regarding the methodological approach, and regarding the metric used to quantify "climate impact".

As such, I believe that major revisions are needed before this paper is ready for publication in ACP. I list major concerns below first, followed by minor issues.

Thank you very much for your review and seeing the value of our work to the community. Your input helped to improve the manuscript significantly. In order to provide a higher consistency within this work, we adapted major parts of the manuscript. Please find below an overview on the changes, based on the referees' comments, which will be discussed in more detail further below:

1. Abstract: The abstract was rephrased to better represent the work and results of this work
2. The introduction was extended to cover the chemical processes and an elaborated discussion why the $O_3$ maximum (w.r.t. time and magnitude) is selected for the analysis
3. The model description now includes a description of the lagrangain model used
4. Section 3 was eliminated since the chemical system is now covered in the introduction. A subsection was added to the results, discussing the variability of the $O_3$ maximum and how typical temporal developments of early, mid and late $O_3$ maxima look like.
5. The altitude analysis was changed to now focus on the mean altitude and not the altitude difference

6. The influence of temperature, $NO_x$ and $HO_x$ in summer and winter are now moved to a single section discussing the $O_3$ production efficiency
7. In order to account for the rapid cycling between OH and $HO_2$, we now analyze the OH recycling probability when discussing the $CH_4$ depletion
8. The discussion now includes a section on how the lagrangain modelling approach influences the results
9. All figures were updated to show the relation for winter and summer and make them visually more appealing

Major comments

While innovative, it is not clear to me that the chemistry and physics simulated within the "tracked" air parcel will provide an estimate of aviation's impacts on the atmosphere which are accurate or consistent with the "parent" EMAC model.

1. Is it validated against any estimates using more conventional techniques? The authors cite Grewe et al (2017c) Section 4.3, but this does not appear to have an explicit comparison of the Lagrangian model's result for aviation impacts compared too ther studies. A direct comparison showing (e.g.) the ozone and methane response in EMAC when performing a conventional simulation of an increment in aviation NOx would be very helpful. Alternatively, if such a comparison is already present in (eg) Grewe et al (2017c), a quantitative evaluation (e.g. "agreement to within X% when simulating aviation emissions") would be helpful. If not already present in Grewe et al2017c I recommend that such an analysis be added to Section 5.

Thank you for pointing this out. The concept used in this study is designed to gain more insights in aviation effects, which conventionally can otherwise not be obtained, limiting the comparability to conventional approaches. However, the following four indicators support the consistency of the modelling approach:

1. The transport scheme is reasonably well established (Brinkop and Jöckel, 2019).
2. The chemical response to a local emission agrees well with earlier findings of Stevenson et al. (2004), who simulated the monthly mean response of ozone and methane to a $NO_x$ pulse and which are very similar to the results from this approach (Grewe et al., 2014, their Fig. 9).
3. The use of a trajectory analysis to interpret either observational data or modelling data is well established.
4. A first verification of the resulting global pattern of the atmospheric sensitivity to a local $NO_x$ emission by comparing to sparsely available literature data was promising (Yin et al., 2018).

We added an elaborated discussion on this matter to the discussion section of our manuscript.

Brinkop, S. and Jöckel, P.: ATTILA 4.0: Lagrangian advective and convective transport of passive tracers within the ECHAM5/MESSy (2.53.0) chemistry–climate model, Geosci. Model Dev., 12, 1991–2008, https://doi.org/10.5194/gmd-12-1991-2019, 2019.

Grewe, V., Frömming, C., Matthes, S., Brinkop, S., Ponater, M., Dietmüller, S., Jöckel, P., Garny, H., Tsati, E., Dahlmann, K., Søvde, O. A., Fuglestvedt, J., Berntsen, T. K., Shine, K. P., Irvine, E. A., Champougny, T., and Hullah, P.: Aircraft routing with minimal climate impact: the REACT4C climate cost function modelling approach (V1.0), Geosci. Model Dev., 7, 175–201, https://doi.org/10.5194/gmd-7-175-2014, 2014.

Stevenson, D. S., Doherty, R. M., Sanderson, M. G., Collins, W. J., Johnson, C. E., and Derwent, R. G.: Radiative forcing from aircraft NOx emissions: Mechanisms and seasonal dependence, Journal of Geophysical Research: Atmospheres, 109, https://doi.org/10.1029/2004JD004759, 2004.

Yin, F., Grewe, V., Frömming, C., & Yamashita, H.: Impact on flight trajectory characteristics when avoiding the formation of persistent contrails for transatlantic flights, Transportation Research Part D: Transport and Environment, 65, 466 – 484, doi: https://doi.org/10.1016/j.trd.2018.09.017, 2018

2. More detail on the Lagrangian model would be helpful. For example, what is the total air mass of the well-mixed box? How is diffusion treated? This is important because of the role of non-linear chemistry (see e.g. Kraabøl et al 2002), which could result in suppressed ozone production when concentrations are very high (i.e. early in the plume's development).

We agree that more information on the Lagrangian model is needed such that the reader understands how the transport processes (identified to be important for the resulting ozone and methane change) are simulated. Therefore, a detailed description of the submodel ATTILA was added to the base model description. Note also that the Lagrangian transport scheme has been evaluated in detail by Brinkop and Jöckel (2019).

Brinkop, S. and Jöckel, P.: ATTILA 4.0: Lagrangian advective and convective transport of passive tracers within the ECHAM5/MESSy (2.53.0) chemistry–climate model, Geosci. Model Dev., 12, 1991–2008, https://doi.org/10.5194/gmd-12-1991-2019, 2019.

The approach used appears to quantify the direct effect that aviation $NO_x$ and $H_2O$ could have on the climate through increases in short-term ozone and decreases in methane, on the basis that both are greenhouse gases. However, one of the major effects of aviation $NO_x$ is a long-term reduction in tropospheric ozone, driven by the methane loss (see e.g. Holmes et al 2001). Is this accounted for here? If I have understood the trajectory-tracking mechanism correctly it does not include feedbacks, so I would not expect it to be accounted

for. If so, it would be useful to include an estimate of the total magnitude of the missing long-term ozone loss.

This effect is known as primary model ozone (PMO). Within REACT4C, PMO was not explicitly simulated, as in most other studies, since this change in ozone occurs far beyond the 90 days simulated. Simulating it explicitly would be too computationally expensive and unfeasible for this modelling approach. Instead a constant scaling factor was applied to the climate impact due to changes in methane, to account for PMO. Due to this simplified approach, PMO cannot be taken into account in this study. We added a paragraph to section 2.3 to explain these circumstances.

The analysis is predicated on the idea that the time and magnitude of the peak change in ozone is a significant indicator of overall climate impacts. However, it is not clear to me that this is the case, and I could not find a quantitative justification or citation to this effect in the paper. The closest I found was the assertion that, in REACT4C, a higher ozone concentration change "generally" leads to a larger RF (p7, line 12). Why use these metrics instead of (e.g.) the total integrated ozone perturbation over the time of the simulation?

In general, we agree that the most obvious variable for addressing the climate impact is the integrated ozone perturbation or the resulting RF. We have realized that our introduction guided both reviewers into a wrong direction on the objective of this paper. Apologies for that. Identifying the resulting climate impact is not the objective of this paper. In this work, we are mainly interested in how transport processes effect the resulting ozone and methane change. By using our dataset, we identify that two characteristics (time and magnitude of the ozone maximum) differ most under varying weather conditions. To improve the introduction and to avoid such a misinterpretation, we added an additional figure (Figure 1 in the revised version), a table (Table 1 in the revised version) and an additional paragraph to the introduction clarifying the purpose of this manuscript. In the figure we show that two emission regions next to each other lead to different ozone perturbations, characterized by a different time and magnitude. Since, the emission occurs under different weather conditions (in and west of a high pressure ridge), we open the question if these differences can be explained by the different weather conditions experienced by the air parcels.

At the end of section 2.1, it is stated that "50 trajectories [are initialized] at 6, 12 and18 UTC", but this is then immediately followed by "in the present study, only 12 UTC is considered". I'm confused – why have the first statement? Also, what is the error which we can expect from only including one time point? Won't this result in same geographical biases?

6, 12 and 18 UTC was only simulated for WP1 within REACT4C. Grewe et al. 2014 demonstrated that the results are more sensitive to the horizontal than the temporal representation. It was thus decided to only simulate 12 UTC for all other weather patterns, due to limited computational resources. We therefore can only include 12 UTC within this study. We updated this paragraph to include this additional information.

Grewe, V., Frömming, C., Matthes, S., Brinkop, S., Ponater, M., Dietmüller, S., Jöckel, P., Garny, H., Tsati, E., Dahlmann, K., Søvde, O. A., Fuglestvedt, J., Berntsen, T. K., Shine, K. P., Irvine, E. A., Champougny, T., and Hullah, P.: Aircraft routing with minimal climate impact: the REACT4C climate cost function modelling approach (V1.0), Geosci. Model Dev., 7, 175–201, https://doi.org/10.5194/gmd-7-175-2014, 2014a.

Many of the conclusions seem to treat correlations as causal links (e.g. section 4.3).Some conclusions – such as that "during summer the $O_3$ formation is limited by the background $NO_x$ concentration, whereas in winter low $HO_2$ concentrations limit the total $O_3$ gained" – do not seem to be sufficiently supported by the data. It would be helpful to see a more explicit justification for why this must be the mechanism.

In order to approach this, we performed major parts of the analysis again. In addition, parts of Section 4.1, 4.2 and 4.3 were merged into a single section (now Section 3.3). In this section we now discuss which factors influence the efficient production of O3. In the case of the $O_3$-$NO_x$-$HO_x$ relation, $NO_x$ and $HO_x$ are analyzed in parallel. The mechanistic why $NO_x$ and $HO_x$ limit the production of $O_3$ in summer and winter was added.

More generally, much of the analysis is not very quantitative – such as page 14, lines 3-5which states that a visual inspection of a correlation makes it "evident" that winter pat-tern 5 looks "almost the same" as for summer pattern 3. I strongly recommend that the authors redo this analysis in a more quantitative fashion and remove conclusions which cannot be both quantified and mechanistically explained in a way which is supported by data.

Thank you very much for pointing this out. In general, we agree with the comment. Some of the analysis is not obvious to the reader, based on the figures provided. We therefore expanded most of the analysis performed in this study and added results for summer and winter. In this particular case, the distribution of $HO_x$ is now presented for WP1, WP5 and SP3 in an additional figure. The reason why $HO_x$ is not the limiting factor for WP5 is now addressed by using this figure and mean and median values.

Some of the analysis used for chemistry is difficult to interpret, and seems to ignore the cycling nature of certain key species. For example, on page 13, it is stated that "35%and 42% of background OH is produced by $HO_2$ reacting with $O_3$ and NO, respectively". However, OH and $HO_2$ are expected to be cycling rapidly. As such, "production" of OH in this fashion is hard to interpret, since it usually matters more to consider what the sources of $HO_x$ are. I am therefore skeptical of the claim that $H_2O$ is only a minor source of OH. I recommend that the authors rephrase this discussion to be about the OH/$HO_2$ ratio and the production and loss of $HO_x$ than to talk separately about OH and $HO_2$.

This is a good point and an important fact missing in our analysis so far. Two measures were performed to account for the cycling between OH and $HO_2$. In Section 3.3 (earlier Section 4.3) the analysis was performed with respect to $HO_x$ instead of only $HO_2$.

Additionally, Section 3.4 (earlier Section 4.4) was completely redone. In order to account for the cycling between both species, we now analyze the OH recycling probability.

Minor issues

P2 l22: I went back and checked the claim that Gilmore et al (2013) showed that "during summer the climate impact is up to 1.5 times higher and only half in winter, when compared to the annual meaning". I do not think this is true. They did show that the ozone production efficiency was 50% higher than the average in summer, but this is largely compensated by changes in ozone lifetime. The overall change in ozone production rate is only about 10% above the annual mean in summer, and correspondingly about 10% below the mean in winter (see Figure 1 of said paper).

This is indeed correct. Thank you for pointing this out. We intended to write about the $O_3$ production efficiency which is about 50% higher in summer. We changed this section in the manuscript to represent their findings appropriately.

The claim that "the standard deviation of background concentrations [of ozone and $NO_x$] are generally considered to be higher than changes induced by aviation . . . making them hardly detectable" (p16 l30, and paraphrased on page 14, lines 17-19) is based on a single, outdated study. Wauben et al 1997 uses a 1995 aircraft $NO_x$ inventory, now 20-25 years old. Since total aviation $NO_x$ emissions have likely more than doubled since then (e.g. Wasiuk et al 2016), I think this claim either needs a more recent and robust backing or it should be removed.

We could not find a more recent peer-reviewed study addressing this problem. Based on their findings we still think that even a doubling of the $NO_x$ emissions will not be detectable on a global scale and that their statement is still valid. We therefore keep this part within the discussion.

The manuscript has several spelling and grammar mistakes. For example, in several locations (e.g. p2, l34) the authors use the word "adopt" when I suspect "adapt" is intended, and on p12 line 19 the word "exited" should be "excited". I would recommend another sweep through the manuscript to fix these and other typos.

The errors pointed out are changed in the manuscript. In addition, the manuscript was checked for further errors of this type.

What happened to WP2 in Table 1?

WP2 was not taken into account in our analysis due to technical problems. These problems lead to an incomplete dataset. We added this information to section 2.3.

---

## Author Response (AR2)

Dear Mr. West,

We are grateful for the reviewers' comments since they were very helpful and we generally agree with the suggested changes. With the new version of the manuscript we are confident that all requests of both reviewers are fulfilled. We are willing to perform further adjustments, if you think that it is necessary.

At this point we would like to thank you for all your work and your help in improving this manuscript!

Kind regards,
On behalf of the authors,

Simon Rosanka

Reply to the Review by William Collins (Referee)

Thank you very much for the helpful comments. Please find in black the original comments from William Collins and in red our replies.

This manuscript is substantially improved, but needs minor revisions still.

I think I understand from the author responses that the main theme of the study is that the time of ozone maximum is a useful diagnostic for the overall effects of aircraft $NO_x$. This needs to be clarified in the text, as it is still not obvious. In particular there are still instances where the text refers to an early ozone maximum leading to or causing some effect. These all need to be more explicit that this is a correlation, not a causal link.

Thank you for pointing this out. All these statements have been adjusted and just mention the correlation now.

The text states that this diagnostic is computationally cheaper than running a full GCM, but doesn't take this any further in explaining how the diagnostic could be used in practice.

This is a good point. We added an elaborate analysis on this to the discussion section of the manuscript. This includes a current state analysis and explains how our findings could be used to allow re-routing of flights on a day to day base. The addition also includes a new table (Table 3).

Abstract, last sentence: This study doesn't (but should) show how the findings can be used to towards a climate impact assessment.

We added an analysis of this topic to the discussion section. Thus, the abstract was not changed.

Page 2, line 19: The discussion of the PMO needs to give a bit more detail.

An elaborate discussion on PMO was added to the manuscript.

Table 1: The caption should state where these regions are: ie. "in a ridge", "to the west of a ridge". PMO needs writing out in full in the caption.

The information on the location was added to the caption of Table 1 and for the definition of PMO a table footnote was added.

Figure 1: The caption needs to state whether the time evolution of the chemical burdens in the lower plot is only along the parcel trajectories or a global change.

Added.

Page 3, line 10: The NOx cycling needs to be explained in a bit more detail. The lifetime of $NO_x$ is only around 2 days and is not washed out. Presumably the $NO_x$ cycles through $NO_y$ reservoirs ($HNO_3$, PAN) and it is the $HNO_3$ that is washed out.

We agree that more information about the involved processes are needed. We added a more detailed explanation of this process.

Page 4, line 11: No, the earlier ozone does not cause a higher integrated $O_3$, it is correlated with it.

Thank you very much for pointing this out. You are indeed correct. Both metrics are correlated but do not cause each other. We updated this sentence and corrected each other occurrence of this claim within the manuscript.

Page 8, line 25: This wording needs to be more precise. Do these really have a "maximum at the end of the simulation" i.e. day 90? Or is it that they don't have a maximum at all i.e. they are still increasing by day 90 – which is what you say a couple of sentences later.

These air parcels have no distinct maximum within the 90 simulated days. At the end of the simulations ozone is still produced for these air parcels. We changed the text accordingly.

Page 8, line 26: How high are these latitudes? Be specific.

These air parcels are mainly emitted north of 50˚N (> 70%). We added this information to the text.

Page 10, line 4: It may be better to be explicit "air parcels with an early $O_3$ maximum are those that are transported to lower latitudes". i.e. the early $O_3$ maximum doesn't cause the transport.

Your proposed wording better represent the findings in this manuscript. We adjusted the text accordingly.

Page 10, line 6: better "air parcels with a late maximum are those that mostly stay …" i.e. the late maximum doesn't cause them to stay.

We agree, see also discussion above. We adapted the text accordingly.

Page 11, line 4: I don't think (subgrid) deep convection affects the trajectories in your analysis.

In principle, the transport scheme ATTILA has the option to deal with subgrid convection and distributes the parcels according to a probability scheme taking updraft downdraft and subsidence into account. However, this option was only operationally available long after the simulations for this work were finished. We deleted the part "in deep convection".

Page 11, line10 – page 12, line 1: The thickness doesn't determine a high pressure system as it is purely temperature. You need to use the geopotential heights to determine the synoptic conditions.

Thank you for pointing this out. We now analyze the 250 hPa geopotential height. All conclusions from this analysis are still valid. The figure and text was adjusted accordingly. The inter-seasonal analysis was adjusted to include the changed analysis (including changes to Table 2).

Figure 5, since thickness and temperature show more or less the same thing you don't need both. I suggest using geopotential height instead for the top plot.

We changed the analysis to use geopotential height (see above).

Page 18, line 22: DU refers to burden, not concentration.

This is correct. The wording was changed.

Page 20, line 6: Better to say that high $O_3$ maxima are only" found", rather than "possible" You have shown a correlation, but not a causation.

Thank you for providing this adjusted wording. We changed it accordingly.

Page 20, line 18-19: "allows" is too strong. You have shown that they are correlated, but not that one causes the other.

Changed.

Reply to the review by the Anonymous Referee #2

Thank you very much for the helpful comments. Please find in black the original comments from reviewer #2 and in red our reply.

I thank the authors for the depth of work they have performed in responding to the reviews. The majority of my concerns have been addressed. The new analysis of $HO_x$ cycling in particular is a significant improvement over the analysis that was previously present.

However, I do still have one significant concern, which remains from my previous review. The paper's focus is on the time and magnitude of the maximum in excess ozone due to an aviation $NO_x$ emission. In the abstract, introduction, and conclusions, the authors use these metrics as a proxy for climate impact. In the response to the review, they state that the introduction now explains why these metrics are chosen and why they are appropriate. The authors also stated that "[i]dentifying the resulting climate impact is not the objective of this paper", which is fair. If indeed the manuscript can be modified to very clearly limit the scope to investigating the influence of weather on only the magnitude and timing of the ozone maximum, then my concerns would be fully addressed.

However, the abstract still states the "the controlling factor to identify the climate impact from aviation NOx emissions are transport processes", which I do not think is supported. The finding that the timing and magnitude of the ozone peak will both change in response to weather patterns is interesting indeed. However I do not think that the paper yet provides sufficient evidence that a larger, earlier peak in the ozone perturbation inevitably means a larger long-term ozone perturbation, and therefore a larger climate impact. In particular I do not agree with the statement on page 4, line 11 that "It becomes obvious that an earlier and larger $O_3$ change leads to a higher integrated $O_3$ and a higher climate impact". This statement seems to be contradicted by Figure 3, which shows that a delayed ozone maximum (the "late" case) could result in a significant integrated impact, depending on the evolution after the 90 day point. The authors direct the reader to Frömming et al 2020 and Grewe et al 2014 in defense of their point. However, the former was only submitted to ACPD and as such not yet peer-reviewed, and does not (as of the time of this review) appear to be available as a preprint. The latter paper is a model development paper and does not quantify the relationship between the ozone peak and the long-term climate impact. As such I could not verify these claims.

Thank you very much for drawing our attention to this issue. In the revised manuscript our intension was to clearly separate our analysis from the resulting climate impact. We only used the resulting climate impact to identify characteristics of interest in the temporal development. In order to make this difference clear, we adjusted each of the statements you mentioned. Additionally, the complete manuscript was reviewed for similar statements. The statement "It becomes obvious that an earlier and larger $O_3$ change leads to a higher integrated $O_3$ and a higher climate impact" (page 4, line 11) was changed to "An earlier and larger $O_3$ change correlates with a higher integrated $O_3$ and a higher resulting climate impact". We do not think that this statement contradicts the "late" case of Figure 3, since almost all $NO_x$ is removed at the end of simulation:

[Figure]

When assuming an exponential decay of $O_3$ after 90 days simulated, the resulting climate impact from the "late" case is expected to have a smaller resulting climate impact then the "early" case. We added a sentence covering this to the manuscript.

I recommend that the authors either: 1) quantitatively and rigorously demonstrate that the timing and magnitude of the ozone maximum correlates with a more conventional metric of impact, such as integrated ozone perturbation; or 2) remove claims that aviation's climate impacts are predicted by these metrics. In the latter case I believe that only minor changes would be needed, specifically to moderate some of the impact claims or to make clear the limitations of this approach.

Quantifying point 1) of your recommendations is beyond the scope of this manuscript. Therefore, the claims that the climate impact is controlled by these characteristics are removed. Instead it is only mentioned that they correlate with each other. This is also done to meet the request of the other referee. We now include an analysis in the discussion on how the results of this manuscript could be used in the future to estimate the climate impact from aviation by using only dynamic simulations. However, we clearly state that this is just a proposal and further investigations in other studies are necessary.

Minor comments

There is a contradiction on page 2. On line 25, it is stated that a larger climate impact occurs at low altitudes, but then on line 31 it is stated that climate impacts are generally larger for emissions at high altitudes.

This is indeed correct. Thank you for pointing this out. In Koehler et al. 2013, a higher climate impact is observed for regions with low aviation activity. We now include an explicit comparison between Europe and India.

It also appears that the revisions have introduced a number of new grammatical errors (e.g. the very first line of the introduction: "…climate change has been well established since years and it is well know…"; page 2, line 31 of the marked-up manuscript reads "…by $O_3$ out weights the cooling…"; page 3, line 21 reads "In exact, one emission region is within…"; page 4, line 1 stats "only little $O_3$ is produced", etc). I recommend that authors make a few iterations to clean these up, and thus maximize the impact of what I believe to be an important paper.

We are pleased that the reviewer sees this potential in our manuscript. A well written manuscript is always favorable. The complete manuscript was reviewed, in addition to the grammatical errors you pointed out.

[revised manuscript text omitted]